# Confident-Anchor-Induced Multi-Source-Free Domain Adaptation

**Jiahua Dong[1,2*], Zhen Fang[3*], Anjin Liu[3], Gan Sun[1†], Tongliang Liu[4]**

[1]State Key Laboratory of Robotics, Shenyang Institute of Automation,
Chinese Academy of Sciences.
[2]University of Chinese Academy of Sciences.
[3]DeSI Lab, AAII, University of Technology Sydney. [4]TML Lab, University of Sydney.
{dongjiahua1995, fzjlyt, sungan1412}@gmail.com, anjin.liu@uts.edu.au,
tongliang.liu@sydney.edu.au

## Abstract

Unsupervised domain adaptation has attracted appealing academic attentions by transferring knowledge from labeled source domain to unlabeled target domain. However, most existing methods assume the source data are drawn from a single domain, which cannot be successfully applied to explore complementarily transferable knowledge from multiple source domains with large distribution discrepancies. Moreover, they require access to source data during training, which are inefficient and unpractical due to privacy preservation and memory storage. To address these challenges, we develop a novel Confident-Anchor-induced multi-source-free Domain Adaptation (CAiDA) model, which is a pioneer exploration of knowledge adaptation from multiple source domains to the unlabeled target domain *without any source data, but with only pre-trained source models*. Specifically, a source-specific transferable perception module is proposed to automatically quantify the contributions of the complementary knowledge transferred from multi-source domains to the target domain. To generate pseudo labels for the target domain without access to the source data, we develop a confident-anchor-induced pseudo label generator by constructing a confident anchor group and assigning each unconfident target sample with a semantic-nearest confident anchor. Furthermore, a class-relationship-aware consistency loss is proposed to preserve consistent inter-class relationships by aligning soft confusion matrices across domains. Theoretical analysis answers why multi-source domains are better than a single source domain, and establishes a novel learning bound to show the effectiveness of exploiting multi-source domains. Experiments on several representative datasets illustrate the superiority of our proposed CAiDA model. The code is available at https://github.com/Learning-group123/CAiDA.

## 1 Introduction

Unsupervised Domain Adaptation (UDA) [22, 59, 63] captures transferable knowledge from labeled data in a source domain to classify unlabeled data in a target domain. UDA has achieved remarkable successes in many applications, *e.g.*, object detection [23], medical diagnose [9, 12], sentiment analysis [34], etc. Generally, most of existing methods employ adversarial learning [17] to encourage the learned source and target features to be indistinguishable from each other [10, 15], or minimize the distribution discrepancy across domains by matching the statistical moments of distributions [41].

---

*Equal contributions

†Corresponding author

35th Conference on Neural Information Processing Systems (NeurIPS 2021).

However, the above-mentioned methods have a strong assumption that the source data are merely drawn from a single domain. Unfortunately, the source data are often collected under different deployed environments (*i.e.*, multiple source domains with large distribution discrepancies) in real-world applications, which makes them difficult to explore complementarily transferable knowledge from the multi-source domains for target prediction. To achieve this, Multi-Source Domain Adaptation (MSDA) [32, 34, 61] is proposed to match the features across domains and then quantify the contributions of source domains [2, 41, 60]. Additionally, [29, 58] aim to weight the source contributions by normalizing the distance similarities between source and target domains.

Unfortunately, recent MSDA methods [29, 34, 41, 44] require massive labeled source data when adapting source domains to the target domain. This could make them inefficient and unpractical in real-world applications with sensitive information (*e.g.*, medical diagnosis [12] and recommendation system [24]), due to privacy preservation issues, storage and security concerns [50, 51]. To this end, a new challenging and practical problem named *Multi-Source-Free Domain Adaptation* (MSFDA) is researched, which explores transferable knowledge from multiple source domains to target domain *with only pre-trained source models and without access to any source data*. The trivial solutions for tackling MSFDA via using existing single-source-free domain adaptation methods [25, 30, 33, 57] are to adapt each source model individually and simply take an average prediction of source models. However, they cannot explore the contributions of the complementary information transferred from different source domains, due to the lack of source data. Therefore, tackling the MSFDA problem is a challenging but rarely-researched task.

To address the MSFDA problem, we develop a novel Confident-Anchor-induced multi-source-free Domain Adaptation (CAiDA) model, which is a pioneer exploration to capture transferable information from multiple source models to promote target prediction without access to source data. Specifically, a source-specific transferable perception module is designed to calibrate the contributions of the transferability from multiple source domains. We develop a confident-anchor-induced pseudo label generator to mine pseudo labels for the unlabeled target data, by incorporating with the quantified source transferability contributions. We construct a confident anchor group to assign each target sample with a semantic-nearest confident anchor, and perform feature augmentation between them to generate confident target pseudo label. A class-relationship-aware consistency loss is proposed to ensure the semantic consistency of underlying inter-class relationships across domains via the alignment of soft confusion matrices. Furthermore, based on some mild assumptions, theoretical analysis guarantees that multiple source models could help generate more reliable pseudo labels. Our theoretical analysis also provides a novel learning bound for MSFDA, which reveals that multiple source models help achieve a tighter generalization error bound for the target domain. We verify the effectiveness of our proposed model via comparison experiments on benchmark datasets. The main contributions of our work are summarized as follows:

• We develop a novel Confident-Anchor-induced multi-source-free Domain Adaptation (CAiDA) model to explore transferable knowledge from multiple source domains to assist target prediction with pre-trained source models and without access to source data. To our best knowledge, this paper is a pioneer exploration of multi-source-free domain adaptation in the field of transfer learning.

• We propose a novel MSFDA theory, which shows that multiple source models could improve the possibility of obtaining more reliable pseudo labels under some mild assumptions. Our theoretical analysis also provides a novel generalization bound for MSFDA to show the effect of multiple source models. This novel bound implies a positive answer to the solvability of MSFDA problem.

• A source-specific transferable perception module and a class-relationship-aware consistency loss are designed to quantify the contributions of the transferability of multiple source domains and ensure the semantic consistency of underlying inter-class relationships across domains, respectively.

• Based on the confident pseudo labeling strategy in theoretical analysis, a confident-anchor-induced pseudo label generator is proposed to generate pseudo labels for the target domain by establishing a confident anchor group and assigning each target sample with a semantic-nearest confident anchor.

## 2 Related Work

**Unsupervised Domain Adaptation** aims to borrow transferable knowledge from source domain to promote the prediction of the unlabeled target domain. After Hoffman *et al.* [22] introduce ad-

versarial learning [17] into domain adaptation, enormous adversarial-based methods [8, 11, 15, 59] are proposed to perform feature-level or pixel-level distribution alignment. Besides, some moment matching-based methods [14, 35, 41] focus on matching the distribution statistical moments at different orders to minimize the distribution discrepancy across domains. Furthermore, some researches design the adversarial dropout [28], batch normalization [52] and auxiliary reconstruction tasks [5,16] to narrow the domain discrepancy. Unfortunately, the methods mentioned above assume massive labeled source data are available. This is unpractical due to privacy and security concerns.

**Source-Free Domain Adaptation** (SFDA) [33] is studied to tackle the above challenge. A common strategy in SFDA methods is to mine the confident pseudo labels for target domain. To this end, [33] uses a self-supervised pseudo labeling strategy, and [25] designs a confidence-based sample filtering method. [42] alleviates the negative transfer brought by noisy pseudo labels through confidence reweighting and regularization. In addition to the pseudo labeling strategy, the model adaptation strategy has also been studied. For example, [30, 57] employ an adversarial learning strategy to perform model adaptation with only pre-trained source models. However, they cannot be applied to tackle the MSFDA problem, due to the distribution discrepancies across different source domains.

**Multi-Source Domain Adaptation** is extended from vanilla domain adaptation [12, 15, 41] by exploring transferable knowledge from multiple sources. To capture the relationship between different source domains and a given target domain, Guo *et al.* [19] design a mixture-of-experts model for unsupervised domain adaptation from multiple sources. [56] focuses on determining which source domain is the best for target prediction via dynamic curriculum learning. Some discrepancy-based methods aim to narrow the distribution discrepancy across domains by minimizing different measures such as the *Rényi*-divergence [21] and maximum mean discrepancy [19]. Moreover, some adversarial-based methods focus on optimizing the $\mathcal{H}$-divergence [41, 60], generative adversarial loss [55,61] and Wasserstein distance [32] to make features from multiple sources indistinguishable for a shared discriminator. [34, 44] perform knowledge adaptation at the pixel-level by replaying multiple source domains. Due to lack of source data, above strategies in MSDA may be invalid and unsuitable to address the challenging MSFDA problem. To this end, Ahmed *et al.* [1] utilize nearest distance measure to mine target pseudo labels, and weight the predictions from multiple source models for the MSFDA task. It may result in that the generation process has high probability to obtain noisy labels [4, 31, 54] when the strategy is not matched with the target data, while our model could generate confident pseudo labels from two different perspectives, *i.e.*, geometry and probability.

## 3  Problem Setting

Let $\mathcal{X}$ and $\mathcal{Y} = [K] := \{1, ..., K\}$ denote the feature space and label space. A domain is a joint distribution $P_{XY}$ on $\mathcal{X} \times \mathcal{Y}$. There are $n$ source domains $\{P_{XY}^i\}_{i=1}^n$. For any source domain $P_{XY}^i$, a corresponding neural-network-based predictor (model) $\boldsymbol{h}^i : \mathcal{X} \to \mathbb{R}^K$ is given. Given a target domain $P_{XY}^t$ with unlabeled target data $T = \{\mathbf{x}^j\}_{j=1}^m \sim P_X^t$, i.i.d., the aim of *multi-source-free domain adaptation* (MSFDA) is to classify the unlabeled target data by utilizing $T$ and $\{\boldsymbol{h}^i\}_{i=1}^n$.

Let $\ell$ denote a non-negative loss function defined over $\mathbb{R}^K \times \mathbb{R}^K$. Given a hypothesis space $\mathcal{H} \subset \{\boldsymbol{h} : \mathcal{X} \to \mathbb{R}^K\}$, we denote $\mathcal{L}_s^i(\boldsymbol{h}) = \mathbb{E}_{(\mathbf{x},y)\sim P_{XY}^i} \ell(\boldsymbol{h}(\mathbf{x}), \boldsymbol{\Phi}(y))$ and $\mathcal{L}_t(\boldsymbol{h}) = \mathbb{E}_{(\mathbf{x},y)\sim P_{XY}^t} \ell(\boldsymbol{h}(\mathbf{x}), \boldsymbol{\Phi}(y))$ as the risks with respect to the $i$-th source domain and a given target domain, where $\boldsymbol{\Phi} : \mathcal{Y} \to \mathbb{R}^K$ maps any label $y$ to a corresponding one-hot vector.

The source predictor $\boldsymbol{h}^i$ is a vector-valued function, *i.e.*, $\boldsymbol{h}^i(\mathbf{x}) = [h_1^i(\mathbf{x}), ..., h_K^i(\mathbf{x})]^\top$, and consists of two basic components: feature extractor $\mathbf{f}^i : \mathcal{X} \to \mathbb{R}^d$ and classifier $\mathbf{c}^i : \mathbb{R}^d \to \mathbb{R}^K$, where $d$ is the dimension of extracted features. Therefore, $\boldsymbol{h}^i$ can be rewritten as $\mathbf{c}^i \circ \mathbf{f}^i$. After using softmax as the activation function in the output layer, we have $h_k^i \geq 0$ and $\sum_{k=1}^K h_k^i = 1$. To ensure that each predictor $\boldsymbol{h}^i$ is a relatively accurate predictor for the domain $P_{XY}^i$, we assume that the predictor $\boldsymbol{h}^i$ is $\epsilon$-accurate under $\ell_1$ loss, *i.e.*, $\mathbb{E}_{(\mathbf{x},y)\sim P_{XY}^i} \|\boldsymbol{h}^i(\mathbf{x}) - \boldsymbol{\Phi}(y)\|_{\ell_1} < \epsilon$, for any $i \in [n]$.

## 4  Theoretical Analysis

Without any relations between source domains and target domain, MSFDA cannot be effectively addressed from the theoretical view. To bridge source and target domains, Mansour *et al.* [40] and

Miraj Ahmed *et al.* [1] assume that $P_{XY} = \sum_{i=1}^n \lambda_i P_{XY}^i$ and $P_{Y|X}^t = P_{Y|X}^i = P_{Y|X}^j$, for any $i,j \in [n]$, where $\lambda_i \geq 0$ and $\sum_{i=1}^n \lambda_i = 1$. This assumption is very strong and may be unrealistic in many real-world applications. Motivated by the meta learning [43] and domain generalization [3], in this paper, we propose some novel and mild assumptions to address MSFDA problem.

**Assumption 1 (Meta Assumption.)** *$P_{XY}^t, P_{XY}^1, ..., P_{XY}^n$ are drawn (i.i.d.) from a meta distribution $\mathcal{P}$, which is defined over a joint distribution space $\mathscr{P}_{XY}$.*

**Assumption 2 (Regular Domain.)** *Let the joint distribution space $\mathscr{P}_{XY}$ is endowed with total variation distance $d_{\mathrm{TV}}(\cdot,\cdot)$. The target domain $P_{XY}^t$ is a regular domain, i.e., for any $\sigma > 0$, $\mathscr{P}(\mathcal{N}_{P_{XY}^t}^\sigma) > 0$, where $\mathcal{N}_{P_{XY}^t}^\sigma = \{P : d_{\mathrm{TV}}(P, P_{XY}^t) < \sigma\}$.*

It is easy to check, if the meta distribution $\mathcal{P}$ is discrete or continuous with continuous density function, then the target domain $P_{XY}^t$ drawn by $\mathcal{P}$ is a regular domain with probability 1. In addition, to weaken the assumption $P_{Y|X}^t = P_{Y|X}^i = P_{Y|X}^j$, for any $i,j \in [n]$, our key strategy is to consider the anchor point assumption that is studied in label-noise learning [36].

We say a given point $\mathbf{x}$ is a $\tau$-*anchor point*, if there exists a predictor $\boldsymbol{h}^i$ with the largest score $h_k^i(\mathbf{x})$, such that $h_k^i(\mathbf{x}) - h_c^i(\mathbf{x}) \geq \tau$, for any $c \in [K]$ and $c \neq k$. $\boldsymbol{h}^i$ is the $\tau$-*anchor predictor* for $\mathbf{x}$.

**Assumption 3 ($\tau$-Anchor Point Assumption.)** *Given a $\tau$-anchor point $\mathbf{x}$, suppose that all $\tau$-anchor predictors for $\mathbf{x}$ are $\boldsymbol{h}^{i_1}, ..., \boldsymbol{h}^{i_l}$, then the true label of the $\tau$-anchor point $\mathbf{x}$ is $k$, if $h_k^j(\mathbf{x})$ is the largest score among scores $h_c^{i_1}, ..., h_c^{i_l}$, for any $c \in [K]$.*

When the source predictors are accurate enough, Assumption 3 implies that the source and target conditional distributions are similar in the high confidence region. Hence, it is much weaker than the traditional assumption $P_{Y|X}^t = P_{Y|X}^i = P_{Y|X}^j$, for $i,j \in [n]$. $\tau$ is regarded as a threshold to distinguish which data has highly confident prediction. In general, $\tau$ is close to 1.

Given any $\boldsymbol{h}^i$, it is easy to check that there exists a matrix function $\mathbf{A}^i(\mathbf{x}) = [a_{kl}^i(\mathbf{x})]$ such that $h_k^i(\mathbf{x}) = \sum_{l=1}^K a_{kl}^i(\mathbf{x}) P_{Y|X}^t(l|\mathbf{x})$ with $\sum_{k=1}^K a_{kl}^i = 1$. We say the diagonal elements $\{a_{ll}^i\}_{l=1}^K$ as the *transfer factor* for $\boldsymbol{h}^i$ and $P_{Y|X}^t$. Generally, $\mathbf{A}^i(\mathbf{x}) = [a_{kl}^i(\mathbf{x})]$ is not unique, thus the transfer factor may be not unique. The following theorem indicates that Assumption 3 holds if we give proper assumptions for transfer factor.

**Theorem 1** *Suppose that the Bayesian label is true label [7]. If there exist transfer factors and a constant $B < K$ such that $\max_{i \in [n], l \in [K]} a_{ll}^i \geq B$, then Assumption 3 holds with $\tau > 1 - B/K$.*

Theorem 1 provides a theoretical support for Assumption 3 and indicates that when the transfer factors are positive, Assumption 3 always holds with a proper $\tau$. To further study the highly confident pseudo labeling strategy, the following theorem provides a lower bound to estimate the number of highly confident pseudo labels, *i.e.*, the number of $\tau$-anchor points.

**Theorem 2** *Assume Assumptions 1 and 2 hold and the conditional distribution $P_{Y|X}^t$ can be presented as a labeling function, i.e., $P_{Y|X}^t(y|\mathbf{x}) = 0$ or $1$. Given $\eta > 0$, if $m \geq n$ and $(1-\eta)(1-\tau) > \epsilon + 2\sigma + 2\sqrt{\log(2m/\delta)/2m}$, then with probability at least $1 - \delta - (1 - \mathscr{P}(\mathcal{N}_{P_{X_t Y_t}}^\sigma))^n > 0$, at least $\eta m$ target data are $\tau$-anchor points, where $\epsilon$ is the upper bound of the accuracies of source predictors, and $\sigma$ is introduced in Assumption 2.*

Theorem 2 indicates that multi-source predictors improve the probability to obtain more $\tau$-anchor points. To further understand the effect of multi-source domains, we build a novel learning bound for the MSFDA task. Let $A_\tau$ be the set consisting of all $\tau$-anchor points. Denote the empirical risk $\widehat{\mathcal{L}}_s^\tau(\boldsymbol{h})$ by $\frac{1}{|A_\tau \cap T|} \sum_{\mathbf{x} \in A_\tau \cap T} \ell(\boldsymbol{h}(\mathbf{x}), \Phi(y))$, where $y$ is the label of the anchor point $\mathbf{x}$.

**Theorem 3** *Given Assumption 3 and some assumptions used in Theorem 2, and suppose that the loss $\ell$ has upper bound $M > 0$ and hypothesis space $\mathcal{H}$ has finite Natarajan dimension, for $\eta > 0$, if $m \geq n$ and $(1-\eta)(1-\tau) > \epsilon + 2\sigma + 2\sqrt{\log(2m/\delta)/2m}$, then for any $\boldsymbol{h} \in \mathcal{H}$ and $b \in (0,1)$, there exists a constant $C(b, K)$ such that with the probability at least $1 - 2\delta - 2(1 - \mathscr{P}(\mathcal{N}_{P_{XY}^t}^\sigma))^n$:*

$$\left| \mathcal{L}_t(\boldsymbol{h}) - \widehat{\mathcal{L}}_s^\tau(\boldsymbol{h}) \right| \leq MC(b,K)\sqrt{\frac{\log(2/\delta)}{\eta^{1-b}m^{1-b}}} + M\frac{2\sigma + \epsilon}{1 - \tau - 2\sigma - \epsilon}, \qquad (1)$$

*where $\epsilon$ is the upper bound of accuracies of source predictors, and $\sigma$ is introduced in Assumption 2.*

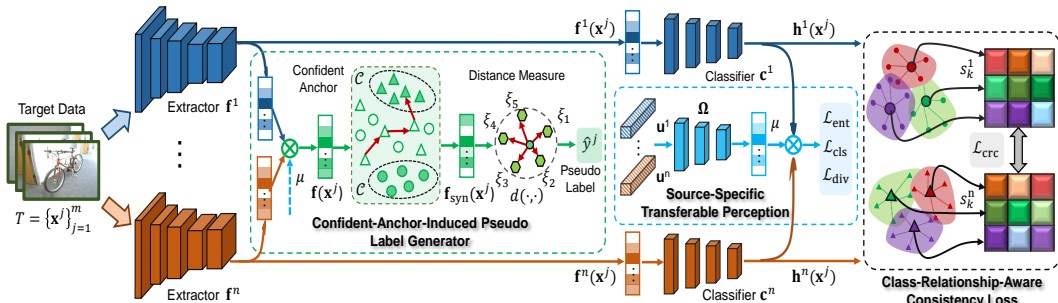

Figure 1: Overview of our model, mainly including a *source-specific transferable perception* strategy to quantify the contributions of the transferability of source domains, a *confident-anchor-induced pseudo label generator* to generate pseudo labels for target domain, and a *class-relationship-aware consistency loss* to ensure the semantic consistency of underlying inter-class relationships.

Theorem 3 shows that multiple source domains improve the probability to ensure a tighter generalization bound, *i.e.*, Eq. (1) holds. Note that the Natarajan dimension used in Theorem 3 is a bit outdated. However, the Natarajan dimension can be replaced and Theorem 3 can be updated without any technical barriers, if there exists better generalization theory for supervised learning.

**Summary of Theoretical Analysis:** The reasons to develop MSFDA are to study the solvability of MSFDA and understand how multi-source predictors benefit the target domain's classification. By Theorems 2 and 3, we realize that multi-source predictors improve the probability to obtain more highly confident pseudo labels, resulting in a tighter generalization bound. The generalization bound in Theorem 3 gives a positive answer to the solvability of MSFDA. Additionally, Theorems 1 and 2 also imply two interesting and important results: Theorem 1 provides the first theoretical support to the confident pseudo labeling strategy, and Theorem 2 provides the first lower bound of the number of highly confident pseudo labels. As we know, the theoretical results in Theorems 1 and 2 are novel.

## 5 The Proposed CAiDA Model

The graphical illustration of our proposed model is depicted in Figure 1. It mainly consists of three significant components: source-specific transferable perception, confident-anchor-induced pseudo label generator and class-relationship-aware consistency loss, which are elaborated as follows.

### 5.1 Source-Specific Transferable Perception

Generally, in multi-source domain adaptation (MSDA), different source domains have different contributions to improve the performance on target domain [21,61]. To this end, many previous MSDA methods match the features across different domains, and then quantify the contributions of source domains by taking the average of the trained source predictors [41,60], or weight the trained source predictors by normalizing the distance similarities [29,58] between source and target domains. However, due to the lack of source data, these methods cannot employ source data to match features and cannot be successfully applied to multi-source-free domain adaptation (MSFDA) tasks.

Therefore, a source-specific transferable perception module is developed to automatically quantify the contributions of the transferability of source domains, as shown in Figure 1. Specifically, the one-hot encoding vector $\mathbf{u}^i \in \mathbb{R}^n$ of the $i$-th source domain can be considered as the unique domain characterization, which is then employed as network input to quantify the contribution of the transferability. We then concatenate all source domains' one-hot characterizations together to obtain $\mathbf{U} = [\mathbf{u}^1, \cdots, \mathbf{u}^n]^\top \in \mathbb{R}^{n \times n}$. $\mathbf{U}$ is then forwarded into a Multi-Layer Perceptron (MLP) network $\mathbf{\Omega}$ to automatically quantify the contribution of the transferability $\boldsymbol{\mu} \in \mathbb{R}^n$ of $n$ source domains: $\boldsymbol{\mu} = \mathbf{\Omega}(\mathbf{U}) = \mathbf{\Omega}([\mathbf{u}^1, \mathbf{u}^2, \cdots, \mathbf{u}^n]^\top)$, such that $\sum_{i=1}^n \mu_i = 1$, where $\mu_i$ denotes the quantified contribution of the $i$-th source domain to the target prediction.

When the source data are unavailable, it is difficult to narrow distribution discrepancy across domains, due to lack of any supervised information. Inspired by [33], we freeze the network parameters of classifiers $\{\mathbf{c}^i\}_{i=1}^n$ and solely perform distribution adaptation across domains on feature extractors $\{\mathbf{f}^i\}_{i=1}^n$ via information maximization [26], since $\{\mathbf{c}^i\}_{i=1}^n$ contain class distribution information of source domains. However, Liang *et al.* [33] cannot be effectively applied to multi-source-free do-

main adaptation scenario, where different source domains have different transferable contributions on target prediction. Therefore, $\mathcal{L}_{\text{ent}}$ is proposed to minimize conditional entropy of target outputs by incorporating the source-specific transferable perception $\mu$:

$$\mathcal{L}_{\text{ent}} = -\frac{1}{m} \sum_{j=1}^{m} [\sum_{k=1}^{K} h_k(\mathbf{x}^j) \log(h_k(\mathbf{x}^j))], \tag{2}$$

where $h_k(\mathbf{x}^j)$ is the $k$-th coordinate value of $\boldsymbol{h}(\mathbf{x}^j)$, here $\boldsymbol{h}(\mathbf{x}^j) = \sum_{i=1}^{n} \mu_i \boldsymbol{h}^i(\mathbf{x}^j)$ denotes the combination of source predictions. The larger value of $\mu_i$ indicates the larger contribution of transferability of the $i$-th source domain for target adaptation. Unfortunately, our proposed CAiDA model may suffer from the trivial solution of Eq. (2) by predicting all target data as a single class to minimize Eq. (2). To tackle this issue, $\mathcal{L}_{\text{div}}$ is designed to consider class prediction diversity by maximizing the entropy of empirical label distribution [6] predicted by different source domains:

$$\mathcal{L}_{\text{div}} = \sum_{k=1}^{K} \hat{p}_k \log \hat{p}_k, \tag{3}$$

where $\hat{p}_k = \frac{1}{m} \sum_{j=1}^{m} h_k(\mathbf{x}^j)$ denotes the mean prediction probability over target data.

## 5.2 Confident-Anchor-Induced Pseudo Label Generator

Although the minimization of both $\mathcal{L}_{\text{ent}}$ and $\mathcal{L}_{\text{div}}$ promotes the class diversity and knowledge adaptation between multiple unseen source data and target data, it cannot circumvent erroneous label assignment due to the noisy target prediction brought by domain discrepancy. To alleviate this issue, based on Assumption 3, a confident-anchor-induced pseudo label generator is developed to mine confident pseudo labels for target data, as shown in Figure 1. Specifically, based on the $\tau$-anchor assumption, we select target data $\mathbf{x}^j$ as confident anchor when the maximum category prediction probability $h_{k^*}(\mathbf{x}^j)$ is larger than the rest of category prediction probabilities $h_k(\mathbf{x}^j)$ ($k \neq k^*$) by a threshold $\tau_p$. For each confident anchor, we integrate the features extracted from multiple source extractors together to obtain a probability-based confident anchor group $\mathcal{C}_p$:

$$\mathcal{C}_p = \{\mathbf{f}(\mathbf{x}^j) | h_{k^*}(\mathbf{x}^j) - h_k(\mathbf{x}^j) \geq \tau_p, \forall k \neq k^*, j = 1, 2, \cdots m\}, \tag{4}$$

where $\mathbf{f}(\mathbf{x}^j) = \sum_{i=1}^{n} \mu_i \mathbf{f}^i(\mathbf{x}^j)$ denotes the feature of the $j$-th target data. $\tau_p$ is defined as the median value of the probability difference between the largest and the second largest probabilities over all target data. Furthermore, inspired by the proposed $\tau$-anchor assumption, to circumvent the noisy anchor in $\mathcal{C}_p$, we construct a distance-based confident anchor group $\mathcal{C}_d$. The feature centroid of the $k$-th class induced by the $i$-th source domain for the whole target data is defined as $\xi_k^i = \sum_{j=1}^{m} h_k^i(\mathbf{x}^j) \mathbf{f}^i(\mathbf{x}^j) / \sum_{j=1}^{m} h_k^i(\mathbf{x}^j)$. $\{\xi_k^i\}_{i=1}^{n}$ are then weighted with source-specific transferability to obtain the feature centroid $\xi_k$ of the $k$-th class over all source domains via $\xi_k = \sum_{i=1}^{n} \mu_i \xi_k^i$. The distance between the feature of $\mathbf{x}^j$ and the $k$-th feature centroid $\xi_k$ is denoted as $d(\mathbf{f}(\mathbf{x}^j), \xi_k)$, where $d(\cdot, \cdot)$ is a distance measure function. When the minimum distance $d(\mathbf{f}(\mathbf{x}^j), \xi_{k^*})$ is shorter than the rest of distances $d(\mathbf{f}(\mathbf{x}^j), \xi_k)(k \neq k^*)$ by a threshold $\tau_d$, we select the target data $\mathbf{x}^j$ into $\mathcal{C}_d$:

$$\mathcal{C}_d = \{\mathbf{f}(\mathbf{x}^j) | d(\mathbf{f}(\mathbf{x}^j), \xi_k) - d(\mathbf{f}(\mathbf{x}^j), \xi_{k^*}) \geq \tau_d, \forall k \neq k^*, j = 1, 2, \cdots m\}, \tag{5}$$

where we set $\tau_d$ as the median value of the distance difference between the shortest and the second shortest distances over all target data. Therefore, the final confident anchor group $\mathcal{C}$ is obtained by performing an intersection operation between $\mathcal{C}_p$ and $\mathcal{C}_d$, *i.e.*, $\mathcal{C} = \mathcal{C}_p \cap \mathcal{C}_d$.

To generate confident pseudo labels for unconfident target data, we select a semantic-nearest confident anchor from $\mathcal{C}$ for each target data via continual similarity searching. To be specific, given an unconfident target data $\mathbf{x}^j$, we search a serial of unconfident guiding data consecutively using the distance measure function $d(\cdot, \cdot)$ in the feature space, until the confident anchor from $\mathcal{C}$ is detected. During each searching process, the previous searched guiding data are not considered in the following iteration. Denote $I$ as the set containing previously searched guiding data, $\mathbf{x}^{jc}$ as the searched confident anchor closest to $\mathbf{x}^j$. The guiding sample $\mathbf{x}^{j(t)}$ in the $t$-th search could be obtained by:

$$\mathbf{x}^{j(t)} = \arg\min_{\mathbf{x}^j \in T} d(\mathbf{f}(\mathbf{x}^{j(t-1)}), \mathbf{f}(\mathbf{x}^j)), \text{ subject to } \mathbf{x}^{j(t)} \neq \mathbf{x}^{j(t-1)}, \mathbf{x}^{j(t)} \notin I, \tag{6}$$

| **Algorithm 1** The Searching Process of Semantic-Nearest Confident Anchor $\mathbf{x}^{jc}$. | **Algorithm 2** The Optimization of Our Model. |
|---|---|
| 1: **Input:** $\mathbf{x}^j, \mathcal{C}, t = 1$; | 1: **Input:** $\{\boldsymbol{h}^i\}_{i=1}^n, T, E$ epoches, $B$ batches; |
| 2: **Initialize:** $\mathbf{x}^{j(t-1)} = \mathbf{x}^j, I = \varnothing$; | 2: **Initialize:** $\kappa_1, \kappa_2$, the parameters of $\mathbf{\Omega}$; |
| 3: **While** $\mathbf{x}^{j(t-1)} \notin \mathcal{C}$ **do** | 3: **For** $e = 1, \cdots, E$ **do** |
| 4:    Obtain $\mathbf{x}^{j(t)}$ via Eq. (6); | 4:    Obtain pseudo labels via Eq. (7); |
| 5:    Update $I$ via adding $\mathbf{x}^{j(t)}$ into $I$; | 5:    **For** $b = 1, \cdots, B$ **do** |
| 6:    Update $\mathbf{x}^{j(t-1)}$ via $\mathbf{x}^{j(t-1)} \leftarrow \mathbf{x}^{j(t)}$; | 6:      Select a mini-batch of target data; |
| 7: **End** | 7:      Update $\{\mathbf{f}^i\}_{i=1}^n$ and $\mathbf{\Omega}$ via Eq.(9); |
| 8: Obtain $\mathbf{x}^{jc}$ via $\mathbf{x}^{jc} \leftarrow \mathbf{x}^{j(t-1)}$ | 8:    **End** |
| 9: **Return** $\mathbf{x}^{jc}$. | 9: **End** |
| | 10: **Return:** $\{\boldsymbol{h}^i\}_{i=1}^n$ and $\mathbf{\Omega}$. |

where $\mathbf{x}^{j(t-1)}$ is the guiding data in the previous search. The searching process of $\mathbf{x}^{jc}$ for the $j$-th target data is summarized in Algorithm 1. Moreover, we fuse the target data and their corresponding confident anchor in the feature space for feature augmentation, and the synthetic feature is denoted as $\mathbf{f}_{\text{syn}}(\mathbf{x}^j) = (1 - \omega)\mathbf{f}(\mathbf{x}^j) + \omega\mathbf{f}(\mathbf{x}^{jc})$, where $\omega \in [0, 1]$ is the random weight to determine the influence of confident anchor on $\mathbf{f}_{\text{syn}}(\mathbf{x}^j)$. Therefore, the confident-anchor-induced pseudo label $\hat{y}^j$ of the $j$-th target data $\mathbf{x}^j$ and the classification loss $\mathcal{L}_{\text{cls}}$ of whole target data are formulated as:

$$\hat{y}^j = \arg\min_{k \in [K]} d(\mathbf{f}_{\text{syn}}(\mathbf{x}^j), \xi_k); \quad \mathcal{L}_{\text{cls}} = \frac{1}{m} \sum_{j=1}^m \sum_{k=1}^K [-\mathbb{1}_{\hat{y}^j = k} \log(h_k(\mathbf{x}^j))]. \tag{7}$$

### 5.3 Class-Relationship-Aware Consistency Loss

The inherent relationships between different classes have semantic consistency across domains, regardless of distribution discrepancy. In light of this, aligning class relationships could promote more shared transferable knowledge from source domains towards target adaptation. To achieve this, as depicted in Figure 1, a class-relationship-aware consistency loss $\mathcal{L}_{\text{crc}}$ is designed to encourage target data from the same class to be compactly clustered together while preserving the intrinsic inter-class relationships via soft confusion matrix alignment. To be specific, the soft label distribution $s_k^i$ of the $k$-th class predicted via the $i$-the source predictor is formulated as $s_k^i = \frac{1}{m} \sum_{j=1}^m [\mathbb{1}_{\hat{y}^j = k} \mu_i h^i(\mathbf{x}^j)]$. The collection of soft label distributions $\{s_k^i\}_{k=1}^K$ represents a kind of soft confusion matrix associated with a particular domain, encoding inter-class relationships learned by the $i$-th source predictor (*e.g.*, computers have more similar semantic relationship with desks than horses). Without access to the source data, $\mathcal{L}_{\text{crc}}$ aims to align soft confusion matrices from different source predictors:

$$\mathcal{L}_{\text{crc}} = \frac{1}{2n^2} \sum_{i=1}^n \sum_{i'=1}^n \sum_{k=1}^K \left( \text{KL}(s_k^i || s_k^{i'}) + \text{KL}(s_k^{i'} || s_k^i) \right), \tag{8}$$

where $\text{KL}(p||q) = \sum_r p_r \log \frac{p_r}{q_r}$ denotes the Kullback-Leibler (KL) divergence. The complexity of $\mathcal{L}_{\text{crc}}$ is not problematic in practice, due to the limited number of source domains.

In summary, the overall optimization objective to optimize $\{\mathbf{f}^i\}_{i=1}^n$ and $\mathbf{\Omega}$ is formulated as:

$$\mathcal{L} = \mathcal{L}_{\text{ent}} + \mathcal{L}_{\text{div}} + \kappa_1 \mathcal{L}_{\text{cls}} + \kappa_2 \mathcal{L}_{\text{crc}}, \tag{9}$$

where $\kappa_1, \kappa_2$ are the balanced weights. The optimization of our model is presented in Algorithm 2.

## 6 Experiments

### 6.1 Datasets and Baseline Methods

**Datasets: Office-31** [22] consists of three representative domains with 31 shared object categories in the office environment, *i.e.*, Amazon (A), Webcam (W) and DSLR (D). **Office-Caltech** [18] is an extension dataset of Office-31 [22] by adding an additional subset called Caltech-256 (C) on it and extracting 10 common object classes among them. **Office-Home** [46] is composed of four different domains including Product (Pr), Clipart (Cl), Art (Ar), and Realworld (Re). Each of these subsets

Table 1: Comparisons between our model and other competing methods on Office-31 [22] dataset (the left block) and Office-Caltech [18] dataset (the right block).

| Methods | Source Data | A, D→W | A, W→D | D, W→A | Avg. | A, D, C→W | A, C, W→D | C, D, W→A | A, D, W→C | Avg. |
|---|---|---|---|---|---|---|---|---|---|---|
| Source only [20] | ✓ | 97.1 | 92.0 | 51.6 | 80.2 | 93.5 | 94.2 | 90.6 | 87.5 | 91.5 |
| MDAN [60] | ✓ | 99.2 | 95.4 | 55.2 | 83.3 | 99.4 | 98.7 | 93.5 | 91.6 | 95.8 |
| DCTN [55] | ✓ | **99.6** | 96.9 | 54.9 | 83.8 | 99.3 | 99.4 | 94.1 | 91.3 | 96.0 |
| M³SDA [41] | ✓ | 99.4 | 96.2 | 55.4 | 83.7 | 99.5 | 99.2 | 94.5 | 92.2 | 96.4 |
| MDDA [62] | ✓ | 99.2 | 97.1 | 56.2 | 84.2 | 99.3 | 99.6 | 95.3 | 92.3 | 96.6 |
| LtC-MSDA [47] | ✓ | **99.6** | 97.2 | 56.9 | 84.6 | 99.4 | 99.7 | 93.7 | 95.1 | 97.0 |
| Source model only | ✗ | 95.4 | 97.5 | 60.2 | 84.4 | 98.0 | 99.5 | 96.3 | 92.1 | 96.5 |
| BAIT [57] | ✗ | 98.5 | 98.8 | 71.1 | 89.5 | 98.0 | 97.5 | 97.5 | 95.7 | 97.2 |
| PrDA [25] | ✗ | 93.8 | 96.7 | 73.2 | 87.9 | 97.6 | 97.1 | 97.3 | 94.6 | 96.7 |
| SHOT [33] | ✗ | 94.9 | 97.8 | 75.0 | 89.3 | 99.6 | 96.8 | 95.7 | 95.8 | 97.0 |
| MA [30] | ✗ | 96.1 | 97.3 | 75.2 | 89.5 | **99.8** | 97.2 | 95.7 | 95.6 | 97.1 |
| DECISION [1] | ✗ | 98.4 | 99.6 | 75.4 | 91.1 | 99.6 | **100.0** | 95.9 | 95.9 | 98.0 |
| Ours-w/oEnt | ✗ | 97.5 | 99.1 | 74.2 | 90.3 | 99.1 | 99.0 | 94.5 | 96.3 | 97.2 |
| Ours-w/oDiv | ✗ | 97.2 | 98.6 | 73.7 | 89.8 | 98.6 | 99.3 | 94.1 | 95.7 | 96.9 |
| Ours-w/oCls | ✗ | 96.7 | 98.4 | 73.0 | 89.4 | 97.3 | 98.4 | 93.6 | 95.2 | 96.1 |
| Ours-w/oCrc | ✗ | 98.6 | 99.5 | 75.4 | 91.2 | 99.6 | **100.0** | 95.3 | 96.5 | 97.9 |
| Ours | ✗ | 98.9 | **99.8** | **75.8** | **91.6** | **99.8** | **100.0** | **96.8** | **97.1** | **98.4** |

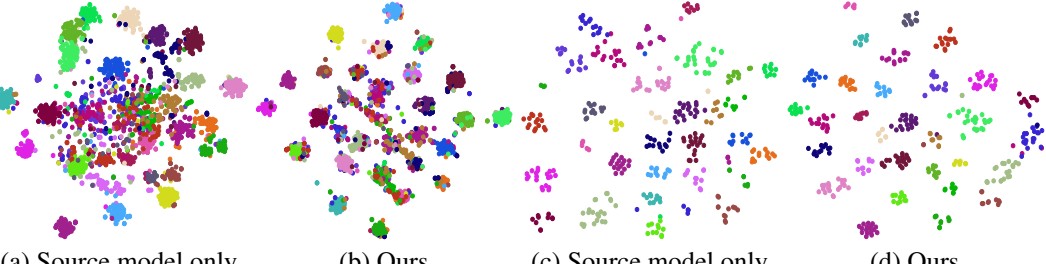

(a) Source model only     (b) Ours     (c) Source model only     (d) Ours

Figure 2: t-SNE [45] visualizations on Office-31 [22] dataset when performing D, W→A (a)(b) and A, W→D (c)(d) domain adaptation tasks.

consists of 65 shared object categories. **Digits-Five** [41] contains five digit recognition subsets including MNIST-M (MM), USPS (UP), MNIST (MT), SVHN (SV) and Synthetic Digits (SY).

**Baseline Methods:** To validate the effectiveness of our model, we conduct comparison experiments with a wide array of baseline methods. Specifically, MDAN [60], DCTN [55], M³SDA [41], MDDA [62] and LtC-MSDA [47] are traditional representative multi-source domain adaptation methods with access to source data. BAIT [57], PrDA [25], SHOT [33] and MA [30] focuses on unsupervised single-source domain adaptation without access to source data. We compare against the multi-source extensions of [25, 30, 33, 57] by taking an average of target soft predictions from all adapted source models. DECISION [1] combines the source adaptation models with suitable weights automatically for multi-source-free domain adaptation. Furthermore, Source only denotes the performance of evaluation on target data when combining the rest of source data for training, and Source model only represents the average performance over the predictions of all source models.

## 6.2 Experiments on Office-31 and Office-Caltech Datasets

**Performance Comparisons:** The comparisons between our CAiDA model and other state-of-the-art methods on Office-31 [22] and Office-Caltech [18] datasets are presented in Table 1. We have the following conclusions from the results in Table 1: 1) When compared with the multi-source domain adaptation methods [41, 47, 55, 60, 62] that employ source data for training, our proposed model without access to source data significantly outperforms them by a large margin of 1.4%~8.3% in terms of mean accuracy. It verifies the superiority of our model to tackle multi-source-free domain adaptation. 2) Our model performs better than the multi-source extensions of single-source-free domain adaptation methods [25, 30, 33, 57], which validates the effectiveness of our pseudo label generation process. 3) The performance of our model is better than [1] for all evaluation tasks, since the confident-anchor-induced pseudo label generator and class-relationship-aware consistency loss promote the adaptation performance. Figure 2 shows that our model significantly narrows distribution discrepancy across domains on Office-31 [22] when compared with Source model only.

**Ablation Studies:** This subsection introduces the effectiveness of each component in our model via ablation studies on Office-31 [22] and Office-Caltech [18] datasets, as shown in Table 1. Ours-

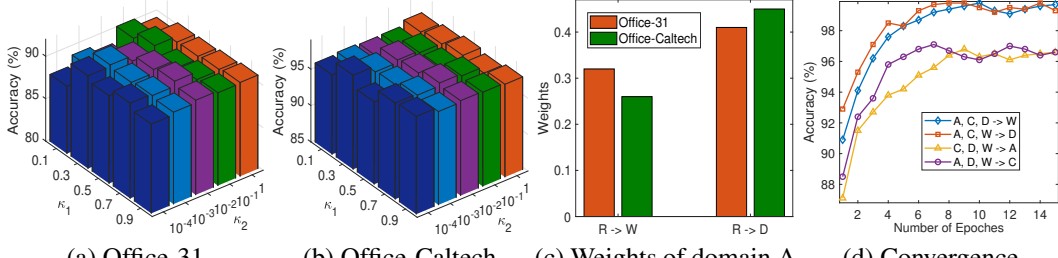

| | (a) Office-31 | (b) Office-Caltech | (c) Weights of domain A | (d) Convergence |

(a) Office-31     (b) Office-Caltech     (c) Weights of domain A     (d) Convergence

Figure 3: Qualitative analysis about parameters $\{\kappa_1, \kappa_2\}$ (a)(b), weights of domain A (c) and convergence on Office-Caltech (d), where R in (c) denotes the rest of domains except for the target.

Table 2: Comparisons between our model and other competing methods on Office-Home [46].

| Methods | Source Data | Ar, Cl, Pr $\rightarrow$ Re | Ar, Cl, Re $\rightarrow$ Pr | Ar, Pr, Re $\rightarrow$ Cl | Cl, Pr, Re $\rightarrow$ Ar | Avg. |
|---|---|---|---|---|---|---|
| Source only [20] | ✓ | 67.8 | 71.3 | 51.8 | 53.4 | 61.1 |
| MDAN [60] | ✓ | 77.3 | 77.6 | 62.2 | 65.4 | 70.6 |
| DCTN [55] | ✓ | 78.7 | 78.3 | 63.8 | 66.4 | 71.8 |
| M$^3$SDA [41] | ✓ | 79.4 | 79.1 | 63.5 | 67.2 | 72.3 |
| MDDA [62] | ✓ | 79.6 | 79.5 | 62.3 | 66.7 | 71.0 |
| LtC-MSDA [47] | ✓ | 80.1 | 79.2 | 64.1 | 67.4 | 72.7 |
| Source model only | ✗ | 76.3 | 78.8 | 50.1 | 50.9 | 64.0 |
| BAIT [57] | ✗ | 77.2 | 79.4 | 59.6 | 71.1 | 71.8 |
| PrDA [25] | ✗ | 76.8 | 79.1 | 57.5 | 69.3 | 70.7 |
| SHOT [33] | ✗ | 82.9 | 82.8 | 59.3 | 72.2 | 74.3 |
| MA [30] | ✗ | 81.7 | 82.3 | 57.4 | 72.5 | 73.5 |
| DECISION [1] | ✗ | 83.6 | 84.4 | 59.4 | 74.5 | 75.5 |
| Ours-w/oEnt | ✗ | 82.6 | 83.0 | 58.7 | 74.2 | 74.6 |
| Ours-w/oDiv | ✗ | 82.1 | 82.9 | 58.5 | 73.8 | 74.3 |
| Ours-w/oCls | ✗ | 81.4 | 82.7 | 57.9 | 73.1 | 73.8 |
| Ours-w/oCrc | ✗ | 83.5 | 84.4 | 59.7 | 74.9 | 75.6 |
| Ours | ✗ | **84.2** | **84.7** | **60.5** | **75.2** | **76.2** |

w/oEnt, Ours-w/oDiv, Ours-w/oCls and Ours-w/oCrc are the abbreviations of training our proposed model without $\mathcal{L}_{\mathrm{ent}}$, $\mathcal{L}_{\mathrm{div}}$, $\mathcal{L}_{\mathrm{cls}}$ and $\mathcal{L}_{\mathrm{crc}}$, respectively. When any one of component of our model is abandoned, the performance degrades 0.4%$\sim$2.3% in terms of average accuracy, which illustrates the rationality and effectiveness of all components to cooperate together. All modules play an indispensable role in improving performance, even though our model has no access to source data.

**Parameter Investigation:** This subsection investigates the effects of hyper-parameters $\kappa_1$ in a range of $\{0.1, 0.3, 0.5, 0.7, 0.9\}$ and $\kappa_2$ in a range of $\{10^{-4}, 10^{-3}, 10^{-2}, 10^{-1}, 1\}$ on Office-31 [22] and Office-Caltech [18] datasets, as shown in Figure 3 (a)(b). It validates that our proposed model achieves stable performance over a wide range of hyper-parameters selection. Moreover, the best performance of our proposed model on target domain is obtained when $\kappa_1 = 0.7$ and $\kappa_2 = 10^{-2}$.

**Contribution Weights and Convergence Analysis:** Figure 3 (c)(d) present the contribution weights of domain A and convergence curves of our proposed model on Office-31 [22] and Office-Caltech [18] datasets. The source-specific transferable perception module could quantify the transferability contributions of complementary information from multiple source domains to target prediction. Furthermore, the accuracy on Office-Caltech [18] dataset converges to a stable value after a few epochs, which demonstrates the convergence effectiveness of our proposed CAiDA model.

### 6.3 Experiments on Office-Home Dataset

**Performance Comparisons:** As presented in Table 2, we conduct comparison experiments on Office-Home [46] dataset to illustrate the effectiveness of our model. We have the following observations from Table 2: 1) Without access to source data, our model significantly outperforms the representative multi-source domain adaptation methods [41, 47, 55, 60, 62] by 3.5%$\sim$5.6% in terms of average accuracy. 2) The confident pseudo label generator encourages our proposed model to perform better than [1], which verifies the superiority of our model for multi-source-free domain adaptation. 3) The performance of our model improves a large margin of 1.9%$\sim$5.5% in terms of mean accuracy, compared with [25, 30, 33, 57]. It validates the efficiency of source-specific transferable perception strategy and class-relationship-aware consistency to narrow distribution discrepancy.

**Ablation Studies:** As introduced in Table 2, we conduct ablation studies of our model on Office-Home [46] dataset to illustrate the rationality of all designed modules. When compared with Ours,

Table 3: Comparisons between our model and other competing methods on Digits-Five [41] dataset, where R denotes the rest of four domains except for the single target domain.

| Methods | Source Data | R → MM | R → MT | R → UP | R → SV | R → SY | Avg. |
|---|---|---|---|---|---|---|---|
| Source only [27] | ✓ | 63.4 | 90.5 | 88.7 | 63.5 | 82.4 | 77.7 |
| MDAN [60] | ✓ | 69.5 | 98.0 | 92.4 | 69.2 | 87.4 | 83.3 |
| DCTN [55] | ✓ | 70.5 | 96.2 | 92.8 | 77.6 | 86.8 | 84.8 |
| M³SDA [41] | ✓ | 72.8 | 98.4 | 96.1 | 81.3 | 89.6 | 87.7 |
| MDDA [62] | ✓ | 78.6 | 98.8 | 93.9 | 79.3 | 89.7 | 88.1 |
| LtC-MSDA [47] | ✓ | 85.6 | 99.0 | 98.3 | 83.2 | 93.0 | 91.8 |
| Source model only | ✓ | 25.2 | 90.0 | 93.3 | 42.8 | 77.8 | 65.8 |
| BAIT [57] | ✗ | 87.6 | 96.2 | 96.7 | 60.6 | 90.5 | 86.3 |
| PrDA [25] | ✗ | 86.2 | 95.4 | 95.8 | 57.4 | 84.8 | 83.9 |
| SHOT [33] | ✗ | 90.4 | 98.9 | 97.7 | 58.3 | 83.9 | 85.8 |
| MA [30] | ✗ | 90.8 | 98.4 | 98.0 | 59.1 | 84.5 | 86.2 |
| DECISION [1] | ✗ | 93.0 | **99.2** | 97.8 | 82.6 | 97.5 | 94.0 |
| Ours-w/oEnt | ✗ | 92.1 | 97.3 | 96.0 | 80.7 | 96.3 | 92.5 |
| Ours-w/oDiv | ✗ | 91.7 | 97.0 | 96.8 | 82.2 | 96.5 | 92.8 |
| Ours-w/oCls | ✗ | 91.3 | 96.6 | 96.4 | 80.5 | 95.8 | 92.1 |
| Ours-w/oCrc | ✗ | 92.8 | 98.2 | 98.1 | 82.8 | 97.7 | 93.9 |
| Ours | ✗ | **93.7** | 99.1 | **98.6** | **83.3** | **98.1** | **94.6** |

the performances of Ours-w/oEnt, Ours-w/oDiv, Ours-w/oCls and Ours-w/oCrc degrades 1.6%, 1.9%, 2.4% and 0.6% in terms of average accuracy, respectively. It validates that all designed modules could cooperate well to address MSFDA task. Moreover, a confident-anchor-induced pseudo label generator could reduce the distribution discrepancy via mining confident pseudo labels.

## 6.4 Experiments on Digits-Five Dataset

This subsection presents the ablation studies and comparison experiments between our model and other competing methods on Digits-Five [41] dataset, as introduced in Table 3. Some conclusions are drawn from the presented results in Table 3: 1) Our proposed model performs the best in terms of average accuracy when compared with multi-source domain adaptation methods [1,41,47,55,60,62] and single-source-free adaptation methods [25,30,33,57]. The significant performance improvement demonstrates the effectiveness of our model, even without access to source data. 2) Our model outperforms [25, 30, 33, 57] by 8.3%∼10.7% mean accuracy, since it automatically quantifies the contributions of different source domains to target adaptation. When we compare our model with [1, 41, 47, 55, 60, 62], the confident-anchor-induced pseudo label generator and class-relationship-aware consistency loss facilitate the performance improvement by mining confident pseudo labels and aligning soft confusion matrices across domains. 3) The performance degradation in ablation studies (*i.e.*, Ours-w/oEnt, Ours-w/oDiv, Ours-w/oCls and Ours-w/oCrc) validates that all proposed components are designed effectively and reasonably to explore transferable knowledge.

## 7 Conclusion and Future Work

This paper proposes a novel confident-anchor-induced multi-source-free domain adaptation (CAiDA) model to capture transferable knowledge from multiple source domains without access to source data. To be specific, a source-specific transferable perception module is developed to automatically weight the contributions of the transferability of source domains. Meanwhile, we design a confident-anchor-induced pseudo label generator to mine confident pseudo labels for the target domain by establishing a confident anchor group, and develop a class-relationship-aware consistency loss to capture consistent inter-class relationships across domains. Theoretical analysis provides some new perspectives to the highly confident pseudo labeling strategy, and gives theoretical support for MSFDA task under some proper mild assumptions. Extensive experiments illustrate the superiority of our proposed model. In the future, we will extend MSFDA to the multi-label [48, 49] or open-set [13, 37, 53] scenarios and use MSFDA techniques to study pandemic [38, 39].

## Acknowledgments

This work was partially supported by National Nature Science Foundation of China under Grant 62003336; National Postdoctoral Innovative Talents Support Program under Grant BX20200353; Nature Foundation of Liaoning Province of China under Grant 2020-KF-11-01; and Australian Research Council Projects under Grant DP-180103424, DE-190101473, and IC-190100031.

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
