# A  Supplementary Material

This supplemental material introduces implementation details, additional comparison experiments, complete proofs and checklist of our proposed model.

## A.1  Implementation Details

**Network Architecture:** Inspired by [33], we utilize a pre-trained ResNet-50 [20] as the feature extractor for object recognition tasks (*i.e.*, Office-31 [22], Office-Caltech [18] and Office-Home [46]). The penultimate fully-connected layer is replaced with a bottleneck layer and a classifier with weight normalization. Batch normalization is employed to normalize the outputs of bottleneck layer. For digit recognition task (*i.e.*, Digits-Five [41]), we utilize a variant of the LeNet [27] as the feature extractor and classifier.

**Training Settings:** Similar to [33], the pre-trained source models are optimized with smooth labels rather than the one-hot encoded labels. It improves the robustness of the trained model by encouraging semantic features to be clustered tightly. For the digit recognition task, the samples from each domain are resized to $32\times32$, and we convert the gray samples to RGB format. The overall framework is trained under an end-to-end manner via back-propagation. The stochastic gradient descent with momentum value as 0.9 is employed as the network optimizer. The initial learning rates for feature extractor and bottleneck layer are respectively set as $10^{-3}$ and $10^{-2}$, while the parameters of classifier are frozen. It is exponentially decayed as the training process. We empirically set the batch size as 64, and set hyper-parameters $\kappa_1$ and $\kappa_2$ as 0.7 and $10^{-2}$. For multi-source-free domain adaptation task, the maximum number of training epoches is set to 15, where the pseudo label generation process occurs at the start of every epoch. All comparison experiments are conducted using Titan XP GPUs with 12 GB memory. For a fair comparison, our proposed model and comparison methods share the same random seed to run the experiments.

Table 4: Comparisons between our model and other competing methods on DomainNet [41] dataset, where R denotes the rest of five domains except for the single target domain.

| Methods | Source Data | R → Cl | R → In | R → Pa | R → Qu | R → Re | R → Sk | Avg. |
|---|---|---|---|---|---|---|---|---|
| Source only [27] | ✓ | 47.6 | 13.0 | 38.1 | 13.3 | 51.9 | 33.7 | 32.9 |
| MDAN [60] | ✓ | 52.4 | 21.3 | 46.9 | 8.6 | 54.9 | 46.5 | 38.4 |
| DCTN [55] | ✓ | 48.6 | 23.5 | 48.8 | 7.2 | 53.5 | 47.3 | 38.2 |
| M³SDA [41] | ✓ | 58.6 | **26.0** | 52.3 | 6.3 | 62.7 | 49.5 | 42.6 |
| MDDA [62] | ✓ | 59.4 | 23.8 | 53.2 | 12.5 | 61.8 | 48.6 | 43.2 |
| LtC-MSDA [47] | ✓ | 63.1 | 28.7 | **56.1** | 16.3 | 66.1 | **53.8** | 47.4 |
| Source model only | ✓ | 49.3 | 14.2 | 39.4 | 12.6 | 53.0 | 35.1 | 33.9 |
| BAIT [57] | ✗ | 57.5 | 22.8 | 54.1 | 14.7 | 64.6 | 49.2 | 43.8 |
| PrDA [25] | ✗ | 57.2 | 23.6 | 55.1 | 16.4 | 65.5 | 47.3 | 44.2 |
| SHOT [33] | ✗ | 58.6 | 25.2 | 55.3 | 15.3 | 70.5 | 52.4 | 46.2 |
| MA [30] | ✗ | 56.8 | 24.3 | 53.5 | 15.7 | 66.3 | 48.1 | 44.1 |
| DECISION [1] | ✗ | 61.5 | 21.6 | 54.6 | 18.9 | 67.5 | 51.0 | 45.9 |
| Ours-w/oEnt | ✗ | 60.9 | 18.7 | 52.6 | 18.6 | 67.7 | 49.4 | 44.7 |
| Ours-w/oDiv | ✗ | 61.5 | 19.2 | 53.1 | 17.6 | 68.4 | 50.6 | 45.1 |
| Ours-w/oCls | ✗ | 60.4 | 18.3 | 52.8 | 16.9 | 67.3 | 50.3 | 44.3 |
| Ours-w/oCrc | ✗ | 62.8 | 21.1 | 53.5 | 18.4 | 70.8 | 51.2 | 46.3 |
| Ours | ✗ | **63.6** | 20.7 | 54.3 | **19.3** | **71.2** | 51.6 | **46.8** |

## A.2  Experiments on DomainNet Dataset

DomainNet [41] is by far the largest and most challenging multi-source domain adaptation dataset. It is composed of around 0.6 million samples and six different domains including Quickdraw (Qu), Clipart (Cl), Painting (Pa), Infograph (In), Sketch (Sk) and Real (Re). Each domain consists of the shared 345 different categories of common objects. There is large distribution discrepancy between any two different domains.

Table 4 reports the comparison experiments and ablation studies of our proposed model on DomainNet [41] dataset. From the introduced performance in Table 4, we could notice that: 1) Our proposed model performs better than prior state-of-the-art comparison methods, when there is no direct access to source data. The source-specific transferable perception strategy effectively quantifies the contributions of each source domain to facilitate the target adaptation performance. 2) The performance of

Table 5: Qualitative analysis about confident anchor group on several benchmark datasets.

| Avg. | Office-31 [22] | Office-Caltech [18] | Office-Home [46] | Digits-Five [41] |
|---|---|---|---|---|
| Ours-w/oCp | 90.5 | 96.9 | 75.1 | 93.7 |
| Ours-w/oCd | 90.3 | 97.2 | 75.3 | 93.3 |
| Ours | **91.6** | **98.4** | **76.2** | **94.6** |

our model degrades 0.5%∼2.5% mean accuracy after removing one of designed components. The ablation studies also illustrate that all components in our proposed CAiDA model play an essential role in narrowing distribution discrepancy across domains for multi-source-free domain adaptation task. 3) The confident-anchor-induced pseudo label generator could mine confident pseudo labels for target data by incorporating with the source-specific transferable perception. It can be effectively illustrated via the degradation performance of Ours-w/oCls and the comparison performance with some representative multi-source domain adaptation methods [41, 47, 55, 60, 62].

### A.3 Qualitative Analysis about Confident Anchor Group

This subsection investigates the effectiveness of our proposed confident anchor group to promote pseudo label generation process, as shown in Table 5. We denote our proposed model without using $\mathcal{C}_p$ and $\mathcal{C}_d$ as Ours-w/oCp and Ours-w/oCd, respectively. When compared with Ours, the performances of Ours-w/oCp and Ours-w/oCd degrade about 0.9%∼1.5%. Such significant performance degradation validates the effectiveness of our proposed CAiDA model to eliminate the noisy pseudo label generation by performing the intersection operation between $\mathcal{C}_p$ and $\mathcal{C}_d$.

### A.4 Proof for Theorem 1.

**Step 1.** According to the condition of Assumption 3 and $\tau > 1 - B/K$, we have that for $\tau$-anchor point $\mathbf{x}$,

$$\max_{i \in [n]} \max_{k \in [K]} h_k^i(\mathbf{x}) > 1 - B/K \geq 1 - \min_{i \in [n]} \min_{l \in [K]} a_{ll}^i(\mathbf{x})/K.$$

**Step 2.** We assume that $h_r^j(\mathbf{x})$ attains the largest value of $h_k^i(\mathbf{x})$ for any $i \in [n], k \in [K]$. Then

$$h_r^j(\mathbf{x}) > 1 - \min_{l \in [K]} a_{ll}^j(\mathbf{x})/K,$$

which implies that for $k \in [K]$ and $k \neq r$,

$$h_k^j(\mathbf{x}) \leq 1 - h_r^j(\mathbf{x}) < \min_{l \in [K]} a_{ll}^j(\mathbf{x})/K \leq a_{kk}^j(\mathbf{x})/K.$$

**Step 3.** First, we note that

$$P_{Y|X}(k|\mathbf{x}) \geq \frac{1}{K} \implies h_k^j(\mathbf{x}) \geq \frac{a_{kk}^j(\mathbf{x})}{K}.$$

Hence, if we assume the Bayesian label is true label, then for $k \in [K]$ and $k \neq r$,

$$h_k^j(\mathbf{x}) < \frac{a_{kk}^j(\mathbf{x})}{K} \implies \mathbf{x} \text{ has label r.}$$

Combining **Step 2** with **Step 3**, we complete the proof.

## A.5 Proof for Theorem 2.

**Step 1.** We claim that if $d_{\mathrm{TV}}(P_{XY}, Q_{XY}) < \sigma$, then $d_{\mathrm{TV}}(P_X, Q_X) < \sigma$.

Given any $g : \mathcal{X} \to \mathbb{R}$ with $|g| \le 1$. We set $f(\mathbf{x}, y) = g(\mathbf{x})$, for any $(\mathbf{x}, y) \in \mathcal{X} \times \mathcal{Y}$, then it is easy to check that
$$|\int f \mathrm{d}P_{XY} - \int f \mathrm{d}Q_{XY}| = |\int g \mathrm{d}P_X - \int g \mathrm{d}Q_X|.$$
Hence, $d_{\mathrm{TV}}(P_X, Q_X) \le d_{\mathrm{TV}}(P_{XY}, Q_{XY}) < \sigma$.

**Step 2.** We claim that if $d_{\mathrm{TV}}(P_{XY}^t, P_{XY}^j) < \sigma$, then
$$\int \|\mathbf{\Phi}(y_1) - \mathbf{\Phi}(y_2)\|_{\ell^1} \mathrm{d}P_{Y|X}^t(y_1|\mathbf{x})\mathrm{d}P_{Y|X}^j(y_2|\mathbf{x})\mathrm{d}P_X^t(\mathbf{x}) < \sigma.$$
First, given any $g : \mathcal{X} \times \mathcal{Y} \to \mathbb{R}$ with $|g| \le 1$, then
$$|\int g \mathrm{d}P_{XY}^t - \int g \mathrm{d}P_{XY}^j| = |\int g \mathrm{d}P_{Y|X}^t \mathrm{d}P_X^t - \int g \mathrm{d}P_{Y|X}^j \mathrm{d}P_X^j|$$
$$\ge |\int g \mathrm{d}P_{Y|X}^t \mathrm{d}P_X^t - \int g \mathrm{d}P_{Y|X}^j \mathrm{d}P_X^t|$$
$$- |\int g \mathrm{d}P_{Y|X}^j \mathrm{d}P_X^t - \int g \mathrm{d}P_{Y|X}^j \mathrm{d}P_X^j|$$
$$\ge |\int g \mathrm{d}P_{Y|X}^t \mathrm{d}P_X^t - \int g \mathrm{d}P_{Y|X}^j \mathrm{d}P_X^t| - \sigma.$$
Now we set $g(\mathbf{x}, y) = \mathrm{sgn}\big(P_{Y|X}^t(y|\mathbf{x}) - P_{Y|X}^j(y|\mathbf{x})\big)$, then
$$\sigma + |\int g \mathrm{d}P_{XY}^t - \int g \mathrm{d}P_{XY}^j| \ge \int \mathrm{d}|P_{Y|X}^t - P_{Y|X}^j|\mathrm{d}P_X^t$$
$$\ge 2 \int \|\mathbf{\Phi}(y_1) - \mathbf{\Phi}(y_2)\|_{\ell^1}\mathrm{d}P_{Y|X}^t(y_1|\mathbf{x})\mathrm{d}P_{Y|X}^j(y_2|\mathbf{x})\mathrm{d}P_X^t(\mathbf{x}).$$
Above inequality has used $P_{Y|X}^t = 1$ or $0$. Hence,
$$\int \|\mathbf{\Phi}(y_1) - \mathbf{\Phi}(y_2)\|_{\ell^1}\mathrm{d}P_{Y|X}^t(y_1|\mathbf{x})\mathrm{d}P_{Y|X}^j(y_2|\mathbf{x})\mathrm{d}P_X^t(\mathbf{x}) < \sigma.$$

**Step 3.** We claim that that if $d_{\mathrm{TV}}(P_{XY}^t, P_{XY}^j) < \sigma$, then
$$\int \big|\|\mathbf{\Phi}(y_1) - \boldsymbol{h}^j(\mathbf{x})\|_{\ell^1} - \|\mathbf{\Phi}(y_2) - \boldsymbol{h}^j(\mathbf{x})\|_{\ell^1}\big|\mathrm{d}P_{Y|X}^t(y_1|\mathbf{x})\mathrm{d}P_{Y|X}^j(y_2|\mathbf{x})\mathrm{d}P_X^t(\mathbf{x}) < \sigma.$$

That is because
$$\int \big|\|\mathbf{\Phi}(y_1) - \boldsymbol{h}^j(\mathbf{x})\|_{\ell^1} - \|\mathbf{\Phi}(y_2) - \boldsymbol{h}^j(\mathbf{x})\|_{\ell^1}\big|\mathrm{d}P_{Y|X}^t(y_1|\mathbf{x})\mathrm{d}P_{Y|X}^j(y_2|\mathbf{x})\mathrm{d}P_X^t(\mathbf{x})$$
$$\le \int \|\mathbf{\Phi}(y_1) - \mathbf{\Phi}(y_2)\|_{\ell^1}\mathrm{d}P_{Y|X}^t(y_1|\mathbf{x})\mathrm{d}P_{Y|X}^j(y_2|\mathbf{x})\mathrm{d}P_X^t(\mathbf{x}) < \sigma.$$

**Step 4.** We claim that if $d_{\mathrm{TV}}(P_{XY}^t, P_{XY}^j) < \sigma$, then
$$\int \|\mathbf{\Phi}(y) - \boldsymbol{h}^j(\mathbf{x})\|_{\ell^1}\mathrm{d}P_{Y|X}^t(y|\mathbf{x})\mathrm{d}P_X^t(\mathbf{x}) < 2\sigma + \epsilon.$$

That is because
$$\int \|\mathbf{\Phi}(y) - \boldsymbol{h}^j(\mathbf{x})\|_{\ell^1}\mathrm{d}P_{Y|X}^t(y|\mathbf{x})\mathrm{d}P_X^t(\mathbf{x})$$
$$\le \int \big|\|\mathbf{\Phi}(y_1) - \boldsymbol{h}^j(\mathbf{x})\|_{\ell^1} - \|\mathbf{\Phi}(y_2) - \boldsymbol{h}^j(\mathbf{x})\|_{\ell^1}\big|\mathrm{d}P_{Y|X}^t(y_1|\mathbf{x})\mathrm{d}P_{Y|X}^j(y_2|\mathbf{x})\mathrm{d}P_X^t(\mathbf{x})$$
$$+ \int \|\mathbf{\Phi}(y_2) - \boldsymbol{h}^j(\mathbf{x})\|_{\ell^1}\mathrm{d}P_{Y|X}^j(y_2|\mathbf{x})\mathrm{d}|P_X^t(\mathbf{x}) - P_X^j(\mathbf{x})|$$
$$+ \int \|\mathbf{\Phi}(y_2) - \boldsymbol{h}^j(\mathbf{x})\|_{\ell^1}\mathrm{d}P_{Y|X}^j(y_2|\mathbf{x})\mathrm{d}P_X^j(\mathbf{x}) < 2\sigma + \epsilon.$$

**Step 5.** According to the finite classes PAC theory, we know if $d_{\mathrm{TV}}(P_{XY}^t, P_{XY}^j) < \sigma$, then with probability at least $1 - \delta > 0$,

$$\frac{1}{m} \sum_{i=1}^{m} \|\boldsymbol{f}(\mathbf{x}^i) - \boldsymbol{h}^j(\mathbf{x}^i)\|_{\ell^1} < 2\sigma + \epsilon + 2\sqrt{\log(2n/\delta)/2m},$$

where $\boldsymbol{f}$ is the true label function of $P_{XY}^t$.

**Step 6.** We still assume that $d_{\mathrm{TV}}(P_{XY}^t, P_{XY}^j) < \sigma$. If at least $(1 - \eta)m$ samples, such that $\|\boldsymbol{h}^j(\mathbf{x}) - \boldsymbol{f}(\mathbf{x})\|_{\ell^1} > t_1 > 0$, then

$$(1 - \eta)t_1 < 2\sigma + \epsilon + 2\sqrt{\frac{\log(2n/\delta)}{2m}}.$$

If we assume

$$(1 - \eta)t_1 \geq 2\sigma + \epsilon + 2\sqrt{\frac{\log(2n/\delta)}{2m}}.$$

Then, we know at least $\eta m$ samples such that $\|\boldsymbol{h}^j(\mathbf{x}) - \boldsymbol{f}(\mathbf{x})\|_{\ell^1} \leq t_1$, which implies that there exists $c \in [K]$ such that

$$h_c^j(\mathbf{x}) - \max_{k \in [K] - \{c\}} h_k^j(\mathbf{x}) \geq \tau,$$

if we set $\tau = 1 - t_1$.

Hence, assume that $d_{\mathrm{TV}}(P_{XY}^t, P_{XY}^j) < \sigma$, if $(1 - \eta)(1 - \tau) \geq 2\sigma + \epsilon + 2\sqrt{\frac{\log(2n/\delta)}{2m}}$, with the probability at least $1 - \delta$, we have at least $\eta m$ samples are the $t$-anchor points.

**Step 7.** With probability at least $1 - (1 - \mathscr{P}(\mathcal{N}_{P_{XY}^t}^\sigma))^n$, there exists $j \in [n]$ such that $d_{\mathrm{TV}}(P_{XY}^t, P_{XY}^j) < \sigma$.

**Step 8.** Combining **Steps 6** and **7**, we complete the proof.

## A.6 Proof for Theorem 3

We denote the distribution generated by all $\tau$-anchor points as $P_X^\tau = P_{X|X \in A_\tau}^t$, where the set $A_\tau$ consists of all $\tau$-anchor points over space $\mathcal{X}$.

**Step 1.** If assume that $m = +\infty$, then according to Theorem 2, we have: with the probability at least $1 - (1 - \mathscr{P}(\mathcal{N}_{P_{XY}^t}^\sigma))^n$, if $(1 - \eta')(1 - \tau) \geq 2\sigma + \epsilon$, then

$$P(A_\tau) \geq \eta',$$

which implies that if we set $(1 - \eta')(1 - \tau) = 2\sigma + \epsilon$, then with the probability at least $1 - (1 - \mathscr{P}(\mathcal{N}_{P_{XY}^t}^\sigma))^n$,

$$P(A_\tau) \geq 1 - \frac{2\sigma + \epsilon}{1 - \tau}.$$

**Step 2.** We consider

$$d_{\mathrm{TV}}(P_X^t, P_X^\tau)$$

Note that

$$d_{\mathrm{TV}}(P_X^t, P_X^\tau) = \sup_A |P_X^t(A) - P_X^\tau(A)|,$$

where $A$ is any measurable set.

Then, with the probability at least $1 - (1 - \mathscr{P}(\mathcal{N}_{P_{XY}^t}^\sigma))^n$, we have

$$
\begin{aligned}
d_{\mathrm{TV}}(P_X^t, P_X^\tau) &= \sup_A |P_X^t(A) - P_X^\tau(A)| \\
&= \sup_A |P_X^t(A) - P_X^t(A \cap A_\tau)/P_X^t(A_\tau)| \\
&\leq P_X^t(A_\tau^c)/P_X^t(A_\tau) \leq \frac{2\sigma + \epsilon}{1 - \tau - 2\sigma - \epsilon},
\end{aligned}
$$

**Step 3.** It is easy to check that

$$\left| \mathcal{L}_t(\boldsymbol{h}) - \int \ell(\boldsymbol{h}(\mathbf{x}), \Phi(y)) \mathrm{d}P_{Y|X}^t(y|\mathbf{x}) \mathrm{d}P_X^\tau(\mathbf{x}) \right| \leq M d_{\mathrm{TV}}(P_X^t, P_X^\tau).$$

**Step 4.** Note that $\mathcal{H}$ has finite Natarajan dimension (we set the dimension is $d$). By Natarajan dimension theory, we know that with the probability at least $1 - \delta$:

$$\left| \widehat{\mathcal{L}}_s^\tau(\boldsymbol{h}) - \int \ell(\boldsymbol{h}(\mathbf{x}), \Phi(y)) \mathrm{d}P_{Y|X}^t(y|\mathbf{x}) \mathrm{d}P_X^\tau(\mathbf{x}) \right| \leq 2M \sqrt{\frac{8d \log \tilde{m} + 16d \log(K + 1) + 2 \log(2/\delta)}{\tilde{m}}},$$

where $\tilde{m} = |A_\tau \cap T|$.

**Step 5.** Combining **Steps 3** and **4**, we have that with the probability at least $1 - \delta$:

$$\left| \widehat{\mathcal{L}}_s^\tau(\boldsymbol{h}) - \mathcal{L}_t(\boldsymbol{h}) \right| \leq 2M \sqrt{\frac{8d \log \tilde{m} + 16d \log(K + 1) + 2 \log(2/\delta)}{\tilde{m}}} + M d_{\mathrm{TV}}(P_X^t, P_X^\tau).$$

**Step 6.** Combining **Steps 5** and **2**, we have that with the probability at least $1 - \delta - (1 - \mathscr{P}(\mathcal{N}_{P_{XY}^t}^\sigma))^n$:

$$\left| \widehat{\mathcal{L}}_s^\tau(\boldsymbol{h}) - \mathcal{L}_t(\boldsymbol{h}) \right| \leq 2M \sqrt{\frac{8d \log \tilde{m} + 16d \log(K + 1) + 2 \log(2/\delta)}{\tilde{m}}} + M \frac{2\sigma + \epsilon}{1 - \tau - 2\sigma - \epsilon}.$$

**Step 7.** According to the results of **Step 6** and Theorem 2, we know that for any $b \in (0, 1)$, there exist a constant $C(b, K)$ such that with the probability at least $1 - 2\delta - 2(1 - \mathscr{P}(\mathcal{N}_{P_{XY}^t}^\sigma))^n$:

$$\left| \widehat{\mathcal{L}}_s^\tau(\boldsymbol{h}) - \mathcal{L}_t(\boldsymbol{h}) \right| \leq M C(b, K) \sqrt{\frac{\log(2/\delta)}{\eta^{1-b} m^{1-b}}} + M \frac{2\sigma + \epsilon}{1 - \tau - 2\sigma - \epsilon}.$$