# OpenReview forum: "Confident Anchor-Induced Multi-Source Free Domain Adaptation"
_NeurIPS.cc/2021/Conference — NeurIPS 2021 Poster_

### Official Review · Reviewer_tYwH · 2021-07-15

**Rating:** 6
**Confidence:** 4

**Summary:**

In this paper, an algorithm for multi-source domain adaptation is proposed. The algorithm is developed in a source-free regime to address the concern for privacy. A loss function is designed to integrate the source-specific end-to-end classifiers using solely the target domain samples, which preserves privacy. The loss function consists of four terms to benefit from enforcing source-specific transferability, anchor-induced pseudo-labels, and class relationship-aware consistency. A PAC-learning style theoretical analysis is provided to demonstrate that using multiple sources may improve the generalizability on the target domain. Finally, experiments on five standard UDA datasets are provided to demonstrate that the method is effective.

**Limitations And Societal Impact:**

I think the authors have addressed these aspects adequately.

**Main Review:**

1. The proofs for both theorems seem correct. However, I feel the analysis does not offer much benefit to the paper. When it comes to the assumption, note:
- the loss function usually is not bounded in deep learning, e.g., using cross-entropy.-  PAC-learning analysis is not very useful for deep neural networks because they are known to have poor VC-dimension or Natarajan dimension. Indeed, this is a primary reason that deep learning cannot be well understood within the standard PAC-learning analysis based on notions such as VC-dimension.- It is not clear to me that the used terms in the loss function satisfy the requirements for Theorem 2.
Hence the provided analysis does not offer much support when deep neural networks are used. Additionally, analyses based on PAC-learning have existential nature broadly speaking and beyond deep learning, you need to have an idea of whether the conditions are met in practice. In this sense, I don't think the theoretical analysis is very valuable. I highly am skeptical to use mathematical analysis when it does not provide much benefit, just for the sake of making the work more rigorous. My assessment would be totally different of course, if it is demonstrated that the analysis is relevant to the experiments.

2. The proposed method outperforms the multi-source UDA methods, despite being a source-free method. This is impressive but goes against my intuition. Because other multi-source UDA methods seem to upper-bounds for the proposed method because they are using more information, i.e., the source samples. I was wondering what is the reason for this observation? Is there any way to show that if the source data points for accessible, the proposed method can have even better performances? I think such an experiment can serve both as an ablative experiment and also broaden the novelty of your work even in simple multi-source UDA setting.

3. The small improvement of the proposed method over the single-source UDA algorithms questions necessity of the proposed approach, i.e., improvement over DECISION is about 0.5% on average. If this is the case, one might ask why not just use the existing source-free methods by averaging? I understand we see a marginal improvement, but this also comes with a significantly more rigorous procedure. Additionally, not all source-free methods are used for comparison. It may be possible that if we use other methods, e.g., Universal Source-Free Domain Adaptation, we get better results. So, at least all the existing source-free methods should be used to demonstrate the proposed method offers an advantage in terms of performance. I would like to include the best source-free single domain UDA algorithm in the results. Please check the literature for this purpose.

4. The primary concern that I have for this work is selecting the values for the hyper-parameters k1 and k2. Figure 3 (a) and (b) are not accurately interpretable in my eyes but I think they show performance can vary a few percent when these hyper-parameters are not tuned properly. Hence, proper value selection for these hyper-parameters seems to be essential to observe the reported performances. Now, the question is how do you tune these parameters? From the descriptions, it seems that the authors have used performance on the target domain to select the optimal values. This is however mathematically wrong because it is as if test error has been used for hyper-parameters tuning. Using the test error for hyper-parameter tuning questions whether the proposed method can stably outperform the baselines given the small differences between the performance, as low as 0.3% in some cases. Please clarify how have you tuned the hyper-parameters in your results and how should the user tune them for other new datasets. If the test error has been used, then the reported performances are not acceptable.

5. I would like to know how does Algorithm 1 scale when we have many source domains? Is it possible to offer complexity analysis? I think theoretical analysis on direct aspects of the algorithm will be more useful.


Minor comments

- The title is confusing: in the absence of punctuation, multi-source free is confusing. Maybe it is better to change it to multisource-free or use different wordings.

- Line 38-40 are a little confusing. It is implied having a massive labeled source of data is the concern for the current MSDA methods, whereas privacy is the issue you address. Irrespective of implementing UDA in a source-free regime, we will need a labeled source domain data.

- Figure 2 is difficult to decode, e.g., what are the colors for each domain?. I understand you have a page limit, but it can be clarified in the Appendix.

----------------------------------
Following internal discussions with the rest of reviewers, I finalized and maintained my updated rating.

**Time Spent Reviewing:**

2

---

> ### Author Response · Authors · 2021-08-09
> **Response to Reviewer tYwH (Q1--Q2)**
>
> Q1:  Questions about theoretical analysis. When it comes to the assumption, note: the loss function usually is not bounded in deep learning, e.g., using cross-entropy.- PAC-learning analysis is not very useful for deep neural networks because they are known to have poor VC-dimension or Natarajan dimension. It is not clear to me that the used terms in the loss function satisfy the requirements for Theorem 2. Additionally, analyses based on PAC-learning have existential nature broadly speaking and beyond deep learning, you need to have an idea of whether the conditions are met in practice.
>
> A1: Thanks for your constructive comments. It is a good question.
>
> 1. The aim of our theory is to explore the solvability of our setting that whether our problem can be solved by finite data under suitable assumptions. Source free domain adaptation (SFDA) has been proposed since 2018 and has became a hot field in transfer learning. It is obvious that SFDA cannot be solved without any assumptions. And few work has shown a theory to discuss the solvability of SFDA. If no papers try to find suitable assumptions to guarantee SFDA is solvable, how can we use SFDA in practice? Our paper is the first work to discuss the solvability of SFDA based on a basic strategy, i.e., confident-based pseudo label strategy  (common used in SFDA). During our researching process, we discover that multiple source models can help us to address the  solvability of SFDA. Hence, we consider the multi-source free domain adaptation problem. In fact, our main theoretical contribution is Theorem 1, which shows that more source models will guarantee more pseudo labels are confident. This results are very meaningful. Because almost all SFDA methods use the confident-based pseudo label strategy as the basic strategy. Hence, Theorem 1 implies that it is more reasonable to consider multiple source than single source. Combining Theorem 1 with empirical risk minimization, Theorem 2 gives a positive answer for the solvability.
>
> 2. We prove the cross-entropy is upper bounded in our algorithm (if we use continuous activating function and the input $\mathbf{x}$ is from a bounded feature space). Let $\mathbf{W}$ be the weight parameters in our neural networks. First, in our code, we have used the F-norm for $\mathbf{W}$ as the regularization (default in pytorch). We also used the softmax in the last layer of our networks. Let $f(\mathbf{x},\mathbf{y},\mathbf{W})$ = $c(h(\mathbf{W},\mathbf{x}),\mathbf{y})$, where $c$ is the cross-entropy loss, $h(\mathbf{W},\mathbf{x})$ is the predicted function for $\mathbf{x}$, and $\mathbf{y}$ is the label. Because we use the softmax, $f$ will  never attain infinity. $f$ is also a continuous function. Because of F-norm for $\mathbf{W}$ as the regularization, we can check $\mathbf{W}$ have upper and lower bounds (i.e., the hypothesis space $\mathcal{H}$ subject to $||\mathbf{W}||_F<C$). If we assume $\mathbf{x}$ is from a bounded feature space, then $f$ is a continuous function defined over a bounded space (because $\mathbf{x, y, W}$ are bounded). This implies that $f$ has upper and lower bounds. Note that in above proof, softmax, F-norm for $\mathbf{W}$ and the  continuity of $f$ are the key to guarantee $f$ has a uniform upper bound.
>
> 3. According to the discussion in 2, Theorem 2 can be extended to a more practice case: $f(\mathbf{x},\mathbf{y},\mathbf{W})$ = $c(h(\mathbf{W},\mathbf{x}),\mathbf{y})$ is a continuous function defined over bounded space.
>
> 4. We need to clarity that the bound of our loss is only necessary to estimate the L1 distance in Step 3 for Theorem 2's proof. The VC-dimension or Natarajan dimension is used in Step 5 for Theorem 2's proof. However, the VC-dimension or Natarajan dimension can be replaced, if there exists better generalization theory for supervised learning. That is by  revising  Step 5 in  Theorem 2's proof, Theorm 2 can be combined with novel generalization theory for supervised learning or deep learning. Hence, we do not think using PAC-learning in Theorem 2 is the main contradiction (we can replace it by better generalization theory). In fact, we think the estimation for lower bound of the number of confident pseudo labels  in Theorem 1 is the key contribution of our theory.
>
> 5. According to your constructive comments, we decide to re-explain our theory, move Theorem 2 to Appendix and give a detailed discussion about the solvability.
>
> Q2. The proposed method outperforms the multi-source UDA methods, despite being a source-free method. What is the reason for this observation? Is there any way to show that if the source data points for accessible, the proposed method can have even better performances? I think such an experiment can serve both as an ablative experiment and also broaden the novelty of your work even in simple multi-source UDA setting.
>
> A2. Thanks for your constructive comments.
>
> 1. The proposed confident anchor-induced pseudo label generator is one of the main reason to make our model outperform the multi-source UDA methods, despite having no access to the source data. Specifically, many multi-source unsupervised domain adaptation methods have designed the pseudo label strategy to narrow distribution discrepancy across domains. However, these methods use pseudo labels roughly, and neglects the fact that there exist many noisy unconfident pseudo labels in the generation process. How to eliminate these  unconfident labels is the key to construct effective pseudo label strategy. To this end, we develop the confident anchor-induced pseudo label generator to mine confident pseudo labels while addressing noisy unconfident labels. Specifically, we first perform the intersection operation between probability-based confident anchor group $\mathcal{C}_p$ and distance-based confident anchor group $\mathcal{C}_d$ to eliminate the noisy pseudo labels in confident anchor group. Then, we use the adjacent areas of the unconfident sample instead of only the unconfident sample to search a semantic-nearest confident anchor. Such similarity searching mechanism improves the generalization ability of pseudo label generation while eliminating the noisy similarity matching. We then use the random weight to fuse the unconfident target data and their corresponding confident anchor in the feature space for feature augmentation and pseudo label generation.  Besides, the proposed source-specific transferability perception module also helps the pseudo label generator to mine confident pseudo labels by weighting the source probability predictions with accurate constributions. The class relationship-aware consistency loss encourages the target data from the same class to be compactly clustered together while preserving the intrinsic inter-class relationships via soft confusion matrix alignment. These two modules are also important to make our model outperform the multi-source UDA methods, despite having no access to the source data.
>
> 3. We further conduct comparison experiments to validate the effectiveness of our model by combining our proposed confident anchor-induced pseudo labeling strategy with several multiple source domain adaptation methods, as shown in the following results. Experiments show our proposed confident anchor-induced pseudo labeling strategy can help these multi-source domain adaptation methods achieve significant improvement about 0.8%$\sim$2.5%. We will update the final version according to your insightful comments.
>
> -------------------------------------------------------------------------------------------------------------------------------------------------------------------------
>
> Avg. (AP) $\qquad\qquad\quad$ | Office-31 $~$ | Office-Caltech | Office-Home | Digits-Five|
>
> DCTN [45] $\qquad\qquad~~$ | $\quad$83.8 $\quad~$ | $\qquad$96.0 $\quad~~~$| $\quad$71.8 $\qquad$ | $\quad$84.8 $\quad$ |
>
> DCTN [45] + Ours $\qquad$| $\quad$86.2 $\quad~$ | $\qquad$97.8 $\quad~~~$| $\quad$74.3 $\qquad$ | $\quad$86.4 $\quad$ |
>
> MDDA [54] $\qquad\qquad~~$| $\quad$84.2 $\quad~$ | $\qquad$96.6 $\quad~~~$| $\quad$71.0 $\qquad$ | $\quad$88.1 $\quad$ |
>
> MDDA [54] + Ours $\quad~~$ | $\quad$86.7 $\quad~$ | $\qquad$97.7 $\quad~~~$| $\quad$73.7 $\qquad$ | $\quad$90.0 $\quad$ |
>
> LtC-MSDA [38] $\qquad~~~~$| $\quad$84.6 $\quad~$ | $\qquad$97.0 $\quad~~~$| $\quad$72.7 $\qquad$ | $\quad$91.8 $\quad$ |
>
> LtC-MSDA [38] + Ours $~~$|$\quad$86.9 $\quad~$ | $\qquad$97.8 $\quad~~~$| $\quad$74.5 $\qquad$ | $\quad$93.2 $\quad$ |
>
> -------------------------------------------------------------------------------------------------------------------------------------------------------------------------

---

> > ### Comment · Reviewer_tYwH · 2021-08-25
> > **Updated Review**
> >
> > Dear authors,
> > Thank you for your extensive response. My concerns are primarily addressed. However, I think at the moment, the updated manuscript is going to be a significantly improved version of the submitted manuscript. As opposed to reviewing the process of journals, I think rebuttals are primarily meant to address unclarity and misunderstandings by the reviewers. Your ablative experiments are of this nature but experiments on more complex datasets are new experiments that could be done at the time of submission. For this reason, I did not improve the score further.

---

> ### Author Response · Authors · 2021-08-09
> **Response to Reviewer tYwH (Q3--Q8)**
>
> Q3: 1) The small improvement of the proposed method over the single-source UDA algorithms questions necessity of the
> proposed approach, i.e., improvement over DECISION is about 0.5% on average. 2) Comparison with other source free method, such as  universal source free domain adaptation.
>
> A3: Thanks for your constructive comments and suggestions.
>
> 1. Compared with DECISION [1], our proposed model achieves slight performance improvement on Office datasets, since these Office datasets are not challenging enough for domain adaptation tasks, and the comparison methods are easy to overfit. To further validate the effectiveness of our model, we conduct comparison experiments on the challenging person re-identification tasks using Market1501 (Ma), DukeMTMCreID (Du), CUHK03 (Cu) and MSMT (Ms) datasets, and set same experimental configuration with Refs[2]. From the following results, we can observe that our model achieves significant performance improvement than multi-source domain adaptation methods (Refs[1] and Refs[2]) with access to source data. Moreover, when compared with source free domain adaptation methods (DECISION [1], SHOT [26] and USFDA Refs[3]), our model also significantly outperforms them by a large margin. It validates the effectiveness of our model to perform the challenging multi-source free domain adaptation tasks.
>
> 2. From the following results, we have experimentally shown that the performance of our model significantly outperforms the universal source free domain adaptation (i..e., USFDA  Refs[3]). In addition, when checking the results of USFDA on Office datasets in USFDA's supplementary materials, we find that the performance of our model on Office datasets is also significantly better than USFDA Refs[3].
>
> Moreover, we have investigated all state-of-the-art methods related to source free domain adaptation, and conducted comparison experiments with them, as shown in the following results and the submitted manuscript. Experiments show our proposed model achieves the state-of-the-art performance when compared with other competing methods.
>
> ------------------------------------------------------------------------------------------------------------------------------------------------------
>
> Metric (mAP // R-1 // R-5) | $~~$ Du+Cu => Ma $\quad$ |  Du+Cu+Ms => Ma $~$| $~~~$ Ma+Cu => Du $~~~$ |
>
> MMT(DBSCAN)   Refs[1]    $~~~$   | 75.3 // 89.5 // 96.6  | 74.8  // 89.3  //  96.2  |  65.7 // 79.0 // 89.0 |
>
> MDIF+RDSBN Refs[2] $\quad~$ | 85.2 // 94.2 // 98.0 | 86.0 // 94.8 // 97.9 | 69.0 // 81.2 // 90.3 |
>
> SHOT [26] $\qquad\qquad\quad~~$ | 81.6 // 92.1 // 94.3 | 81.9 // 91.7 // 92.8 | 64.4 // 76.6 // 84.1 |
>
> USFDA  Refs[3] $\qquad\quad~~~$ | 79.3 // 91.3 // 93.2 | 80.1 // 90.5 // 90.8 | 62.8 // 73.4 // 82.5 |
>
> DECISION [1] $\qquad\qquad~$ | 83.8 // 92.6 // 96.5 | 83.6 // 92.7 // 95.4 | 67.3 // 78.9 // 88.6 |
>
> Ours $\qquad\qquad\qquad\quad~~$ | 86.4 // 95.0 // 98.7 | 87.3 // 96.1 // 98.5 | 69.8 // 82.1 // 91.3 |
>
> ------------------------------------------------------------------------------------------------------------------------------------------------------
>
> Refs[1] Mutual meanteaching: Pseudo label refinery for unsupervised domain adaptation on person re-identification, ICML 2020
>
> Refs[2] Unsupervised Multi-Source Domain Adaptation for Person Re-Identification, CVPR 2021
>
> Refs[3] Universal Source-Free Domain Adaptation, CVPR 2020
>
> Q4: How do you tune the parameters  k1 and k2? Do you have used performance on the test error to select the optimal values.
>
> A4: Thanks for your insightful comments. It is a good question. We do not use test error to select the values for the hyper-parameters in this paper.
>
> In unsupervised domain adaptation (UDA), how to tune the parameters is still a difficult problem. Many existing UDA methods  use the reverse validation error strategy (proposed by ''Cross validation framework to choose amongst models and datasets for transfer learning'' and  "Domain-Adversarial Training of Neural Networks")  to select optimal hyper-parameters. However, this strategy requires to access the source data, which is unavailable in our model (without access to source data). Thus, we consider the consistency of target samples' pseudo labels generated from multiple source models to select the optimal values, since the consistency of pseudo labels from multiple source models could well reflect the degree of performance adaptation. Namely, the better consistency of pseudo labels from multiple source models, the better degree of performance adaptation.
>
> Specifically, we first  randomly select  30%  unlabeled  target  samples, and then measure the consistency between these target samples' pseudo labels generated from individual source model and the pseudo labels mined from our weighting strategy (i.e., Eq (7)). We compute the inconsistency error for each source model, and then weigh all inconsistency errors with source-specific transferability $\mu$ as the corresponding reference error. This reference error can be used to select the optimal values for the hyper-parameters. Thus, we tune the hyper-parameters k1 in a range of \{0.1, 0.3, 0.5, 0.7, 0.9\} and k2 in a range of \{0.0001, 0.001, 0.01, 0.1, 1\} to compute the reference errors. The optimal values of hyper-parameters k1 and k2 can be selected, when their corresponding reference error is minimum. Moreover, we observe that such hyper-parameter selection strategy has close consistent relationships with the experimental results in Figure 3 of our paper, which experimentally validate the effectiveness of this hyper-parameter selection strategy.
>
> Furthermore, our model has stable performance over a wide selection range of hyper-parameters. Thus, when performing experiments on other new datasets, we can use the optimal hyper-parameters of previous experimental dataset, and fine-tune these hyper-parameters via our proposed hyper-parameter selection strategy for new datasets.
>
> Q5: Questions about the complexity analysis and scale of Algorithm 1.
>
> A5: The complexity analysis of Algorithm 1 is $O(nmK+amd)$. Specifically, combining the predictions from n source domains via the contribution weighting strategy to obtain the final target predictions consumes $O(nmK)$, and searching the guiding unconfident samples until the confident anchor costs $O(amd)$, where a=7 is the maximum number of guiding samples in this paper. The complexity of Algorithm 1 increases linearly as the number of source domains, which can efficiently scale to many source domains.
>
> Q6: The title is confusing. Maybe it is better to change it to multisource-free or use different wordings.
>
> A6: Thanks for your valuable suggestions. We will modify the title in the final revision.
>
> Q7: Line 38-40 are a little confusing. It is implied having a massive labeled source of data is the concern for the current MSDA methods, whereas privacy is the issue you address. Irrespective of implementing UDA in a source-free regime, we will need a labeled source domain data.
>
> A7: Thanks for your insightful comments. We will rewrite Line 38-40 for better understanding. In multi-source domain adaptation, the source data from multiple source domains and the target data are available for each other. However, in multiple source free domain adaptation, we can only access the pre-trained source models instead of the source data to achieve the privacy protection.
>
> Q8: Figure 2 is difficult to decode, e.g., what are the colors for each domain?. I understand you have a page limit, but it
> can be clarified in the Appendix.
>
> A8: Thanks for your suggestion. In Figure 2, different colors denote different categories. We will clearly clarify the Figure 2 in the final revision process.

---

### Official Review · Reviewer_R2JJ · 2021-07-16

**Rating:** 6
**Confidence:** 4

**Summary:**

This work presents CAiDA for Multi Source-Free Domain Adaptation in the absence of source data. The core idea is to define confident anchors to obtain pseudo labels for target domains. In the presence of multiple source domains, the contribution of each source domain for the target sample is measured using source-specific transferability perception modules. Further a class-relationship aware consistency loss is proposed to maintain semantic consistency across domains. The proposed method consistently performs better than prior arts on several datasets.

**Limitations And Societal Impact:**

Adequately addressed in the Supplementary Material.

**Main Review:**

Overall, the novelty is limited as the approach ties multiple existing ideas together. However, this work can be improved by focusing more on the distinguishing elements.

1. Novelty:

One of the major drawbacks is that the novelty is limited in the proposed framework, especially when the improvement over prior work DECISION [1] is only marginal (e.g. 0.4% overall improvement on Office-Caltech dataset in Table 1). Specifically, there are two major components which are employed in previous works:

    - The conditional entropy minimization ($\mathcal{L}_{ent}$) and diversity promoting ($\mathcal{L}_{div}$) losses are quite common in the Domain Adaptation literature. For instance, DECISION [1] employs both these losses.

    - While the proposed pseudo-labeling approach is based on the nearest confident anchor (high confidence target sample), DECISION [1] labels target samples based on the nearest cluster centroid. These two methods are notionally similar - both assign target pseudo labels based on the nearest high confidence target samples.

The major distinguishing factor of this work is the Class-Relationship Aware Consistency loss and the synthetic feature augmentation (L234), which should be highlighted upon in my opinion. Also note that certain works such as [P1] optimize class confusion matrices that naturally results in feature alignment during adaptation.

[P1] Jin et al, “Minimum Class Confusion for Versatile Domain Adaptation”, ECCV 2020


2. Readability / Missing Details:

    - For most parts, the manuscript is easy to follow, however there are some inconsistencies in the notation. For eg. the LHS in $\mathcal{L}_t(\boldsymbol(h))$ (L113) should have superscript $i$. In L184, $\mu$ should be bold (vector).

    - Could the authors clarify what is meant by “probability over all target data” in L200? Does it refer to the target data in a mini-batch, or the entire target dataset (L110)? If it is the entire target data (of size $m$) then each backpropagation through loss $\mathcal{L}_{div}$ would require 1 epoch.

    - L183 (“$\boldsymbol{U}$ is then forwarded into a three-layer convolution network $\boldsymbol{\Omega}$”): Could the authors clarify the rationale behind the use of a convolution network? Here, the ordering of the sources in the matrix $\boldsymbol{U}$ would play a major role in determining the performance of the network, due to the nature of the convolution operation. Furthermore, I couldn’t find details about this network.

    - The assumptions in Section 4 are better understood by referring to the detailed proofs in the supplementary material. I suggest making Section 4 more elaborate (for e.g. the discussion in Introduction in L30-L77 is repetitive which can be trimmed down to save space).


3. Approach / Evaluation:

- It seems that the Consistency Loss $\mathcal{L}_{crc}$ (Eq. 8) is computationally intensive, with $O(n^2 * K^2)$ operations. However, from the ablative analysis in Tables 1-3, the consistency loss results in the least gains which suggests that the loss is inefficient. Could the authors comment on this aspect? For instance, what is the increase in training time by incorporating this loss?

- An ablative study on the use of $\mathcal{C}_p$ and $\mathcal{C}_d$ is missing. Note that, using $\mathcal{C}_d$ is similar to using the cluster centroid for pseudo-labeling in DECISION [1]. Therefore, it is important to understand how the combination of $\mathcal{C}_p$ and $\mathcal{C}_d$ helps.

On a side note, it would be a good idea to present standard deviation along with the mean statistics since the average performance of the methods are close.

---------
Post Rebuttal comments: The authors have addressed my major concerns. After going through all the reviews and the author responses, I would like to vote for an acceptance.

**Time Spent Reviewing:**

8

---

> ### Author Response · Authors · 2021-08-09
> **Response to Reviewer R2JJ (Q1--Q5)**
>
> Q1: The improvement over prior work DECISION [1] is only marginal (e.g. 0.4% overall improvement on Office-Caltech dataset in Table 1).
>
> A1: Thank you for the deep analysis of our model. Our proposed model achieves slight performance improvement on Office datasets, since these Office datasets are not challenging enough for recent domain adaptation works and the competing methods are easy to overfit. To further validate the effectiveness of our model, we conduct comparison experiments on the challenging person re-identification tasks using Market1501 (Ma), DukeMTMCreID (Du), CUHK03 (Cu) and MSMT (Ms) datasets, and set same experimental configuration with Refs[2]. From the following results, we can observe that our model achieves significant performance improvement than multi-source domain adaptation methods (Refs[1] and Refs[2]) with access to source data. Moreover, when compared with source free domain adaptation methods (DECISION [1], SHOT [26] and USFDA Refs[3]), our model also significantly outperforms them by a large margin. It validates the effectiveness of our model to perform the challenging multi-source free domain adaptation tasks.
>
> ---------------------------------------------------------------------------------------------------------------------------------------------------------
>
> Metric (mAP // R-1 // R-5) | $~~$ Du+Cu => Ma $\quad$ |  Du+Cu+Ms => Ma $~$| $~~~$ Ma+Cu => Du $~~~$ |
>
> MMT(DBSCAN)   Refs[1]    $~~~$   | 75.3 // 89.5 // 96.6  | 74.8  // 89.3  //  96.2  |  65.7 // 79.0 // 89.0 |
>
> MDIF+RDSBN Refs[2] $\quad~$ | 85.2 // 94.2 // 98.0 | 86.0 // 94.8 // 97.9 | 69.0 // 81.2 // 90.3 |
>
> SHOT [26] $\qquad\qquad\quad~~$ | 81.6 // 92.1 // 94.3 | 81.9 // 91.7 // 92.8 | 64.4 // 76.6 // 84.1 |
>
> USFDA  Refs[3] $\qquad\quad~~~$ | 79.3 // 91.3 // 93.2 | 80.1 // 90.5 // 90.8 | 62.8 // 73.4 // 82.5 |
>
> DECISION [1] $\qquad\qquad~$ | 83.8 // 92.6 // 96.5 | 83.6 // 92.7 // 95.4 | 67.3 // 78.9 // 88.6 |
>
> Ours $\qquad\qquad\qquad\quad~~$ | 86.4 // 95.0 // 98.7 | 87.3 // 96.1 // 98.5 | 69.8 // 82.1 // 91.3 |
>
> -------------------------------------------------------------------------------------------------------------------------------------------------
>
> Refs[1] Mutual meanteaching: Pseudo label refinery for unsupervised domain adaptation on person re-identification, ICML 2020
>
> Refs[2] Unsupervised Multi-Source Domain Adaptation for Person Re-Identification, CVPR 2021
>
> Refs[3] Universal Source-Free Domain Adaptation, CVPR 2020
>
> Q2: There are two major components which are employed in previous works: 1) The conditional entropy minimization and class diversity promoting  losses; and  2) pseudo-labeling approach similar to  DECISION [1].
>
> A2: Many thanks for analyzing our model.
>
> 1. The conditional entropy minimization and diversity promoting losses are two basic loss terms (commonly used in source free methods) to tackle non-adversarial learning based source free domain adaptation task, as introduced in [1][26]. Note that these two losses are not our contributions, and the main contribution in Section 5.1 of this paper is to extend these two losses used in single source free domain adaptation [26] to multi-source free domain adaptation. To this end, we develop the source-specific transferability perception module to automatically quantify the transferability contributions of complementary knowledge from source domains without access to the source data, and linearly combine the probability predictions from all sources as the final predictions for target samples. When compared with DECISION [1] heavily relying on parameter initialization and human prior, our proposed source-specific transferability perception could automatically learn and update the transferability contributions of complementary knowledge from source domains by the network itself and without manual interference (few human prior).
>
> 2. The pseudo-labeling strategy of our model is significant different with DECISION [1]. The pseudo labeling strategy in DECISION [1] utilizes the weighted k-means clustering to compute class-wise prototypes, and measures the nearest distance between the given sample and class-wise prototypes to generate pseudo label for target sample. However, [1] only depends on nearest distance measure to  learn pseudo labels. This may result in that the generation process has high probability to obtain noisy labels when the strategy is not matched with the target data (an unsuitable perspective leads more noisy  labels).
>
> (1) Different from [1], we generate pseudo labels from  two different  perspectives, i.e., geometry and probability. Based on these two perspectives, we develop a confident anchor-induced pseudo label generator to mine confident pseudo labels while addressing noisy unconfident labels. Specifically, we first perform the intersection operation between probability-based confident anchor group $\mathcal{C}_p$ and distance-based confident anchor group $\mathcal{C}_d$ to eliminate the noisy pseudo labels in confident anchor group.
>
> (2) Then, we consider the similarity relationship between a confident anchor and a local unconfident group including multiple semantically-adjacent unconfident samples. It can also be seen that we use the adjacent areas of this unconfident sample instead of only this unconfident sample to search a semantic-nearest confident anchor. We believe using the adjacent areas of unconfident samples will be more robust than only using one unconfident sample to search a semantic-nearest confident anchor. Such similarity searching mechanism improves the generalization ability of pseudo label generation while eliminating the noisy similarity matching.
>
> (3) We use the random weight to fuse the unconfident target data and their corresponding confident anchor in the feature space for feature augmentation and pseudo label generation. This strategy further eliminates the noisy pseudo labels.
>
> Q3: The major distinguishing factor of this work is the Class-Relationship Aware Consistency loss and the synthetic feature augmentation (L234), which should be highlighted upon in my opinion. Also note that certain works such as Refs[1] optimizes class confusion matrices that naturally results in feature alignment during adaptation.
>
> Refs[1]: Jin et al, “Minimum Class Confusion for Versatile Domain Adaptation”, ECCV 2020
>
> A3: Many thanks for this valuable suggestion. The novel contributions of this paper are mainly four parts, instead of only the class relationship-aware consistency loss and the synthetic feature augmentation (L234).
>
> 1. To the best of our knowledge, our paper is the first exploration to propose the theoretical analysis about confident pseudo label generation in transfer learning field. Specifically, theorem 1 has shown that more source domains implies more confident pseudo labels under the confident-based pseudo label strategy (anchor point assumption). Our theorem 1 is the first theorem to explore the relationship between multiple source and pseudo labels by providing a lower bound for the number of confident pseudo labels. As we know, although many works have studied the confident-based pseudo label strategy, no work has provided the lower bound for the number of the confident pseudo labels. This is our core theoretical contributions. In addition, the math skills developed in theorem 1 can also be extended into other fields, such as meta learning theory, domain generalization theory and multiple source domain adaptation theory.
>
> 2. Existing source free domain adaptation methods cannot consider the negative influence of noisy pseudo labels, while our proposed confident anchor-induced pseudo label generator could mine confident pseudo labels and eliminate noisy unconfident labels. Specifically, we first perform the intersection operation between probability-based confident anchor group $\mathcal{C}_p$ and distance-based confident anchor group $\mathcal{C}_d$ to eliminate the noisy pseudo labels in confident anchor group. Then, we use the adjacent areas of the unconfident sample instead of only the unconfident sample to search a semantic-nearest confident anchor. Such similarity searching mechanism improves the generalization ability of pseudo label generation while eliminating the noisy similarity matching, and use the random weight to fuse the unconfident target data and their corresponding confident anchor in the feature space for feature augmentation and pseudo label generation.
>
> 3. Compared with DECISION [1] which heavily relies on the parameter initialization and human prior, the proposed source-specific transferability perception can be initialized with random parameters, and learns the transferability contribution weight of each source domain by the network itself and without manual interference (few human prior).
>
> 4. When compared with Refs[1], our proposed class relationship-aware consistency loss considers the category-wise distribution properties from multiple source domains instead of only one source domain by incorporating with the source-specific transferability contributions. In addition, we use the symmetrical KL divergence instead of Euclidean distance to better measure the intra-class and inter-class relationships across domains.
>
> Q4: Questions about some inconsistencies in the notation (L113, L184).
>
> A4: Thanks for your suggestions. We will carefully polished this paper in the revision.
>
> Q5: Questions about “probability over all target data” in L200.
>
> A5: It refers to the target data in a mini-batch. In the algorithm, we compute this loss $\mathcal{L}_{div}$ via mini-batch samples, when optimizing the network.

---

> ### Author Response · Authors · 2021-08-09
> **Response to Reviewer R2JJ (Q6--Q10)**
>
> Q6: Questions about the three-layer convolution network in L183.
>
> A6: 1) We use the three-layer convolution network to encode source domain characterizations, and learn the transferability contribution of each source domain by the network itself and without any human prior. 2) The ordering of multiple sources has little influence to the final network performance, since the number of input source domains is corresponding to the dimension of the output transferability contributions. 3) It is a three-layer fully-connected network with the channel as \{45, 20, n\} and the learning rate as 0.0001, where $n$ is the number of source domains.
>
> Q7: Questions about making Section 4 more elaborate.
>
> A7: Thanks for your suggestion. We will elaborate Section 4 in the final revision.
>
> Q8: Questions about the consistency loss $\mathcal{L}_{crc}$.
>
> A8: Thanks for your constructive comments.
>
> 1. The consistency loss is more effective when it cooperates with pseudo-labeling strategy. As shown in the following results, the performance of our model degrades about 0.5%$\sim$0.7% mAP when removing the consistence loss. When compared with Ours-w/oCls using probability to compute consistence loss, Ours-w/oCrcCls without using both consistence loss and pseudo-labeling strategy significantly degrades the performance about 3.2%$\sim$3.6%. This performance degradation is significantly large than Ours-w/oCls vs Ours and Ours-w/oCrc vs Ours. It effectively illustrates that the consistency loss $\mathcal{L}_{crc}$ is more effective when cooperating with our proposed pseudo-labeling strategy.
>
> 2. There is a little increase in training time by incorporating the consistency loss $\mathcal{L}_{crc}$, since the number of source domains is usually small, and the maximum number is set as 4 in this paper.
>
> -------------------------------------------------------------------------------------------------------------------------------------------------------------
>
> Avg. (AP) $\qquad$ | Office-31 $~$ | Office-Caltech | Office-Home | Digits-Five|
>
> Ours-w/oCls $~~~$ | $\quad$89.4 $\quad~$ | $\qquad$96.1 $\quad~~~$| $\quad$73.8 $\qquad$ | $\quad$92.1 $\quad$ |
>
> Ours-w/oCrc $~~~$ | $\quad$91.1 $\quad~$ | $\qquad$97.9 $\quad~~~$| $\quad$75.6 $\qquad$ | $\quad$93.9 $\quad$ |
>
> Ours-w/oCrcCls| $\quad$88.4 $\quad~$ | $\qquad$94.8 $\quad~~~$| $\quad$72.7 $\qquad$ | $\quad$91.2 $\quad$ |
>
> Ours $\qquad\quad~~~$ | $\quad$91.6 $\quad~$ | $\qquad$98.4 $\quad~~~$| $\quad$76.2 $\qquad$ | $\quad$94.6 $\quad$ |
>
> -------------------------------------------------------------------------------------------------------------------------------------------------------------
>
> Q9: Questions about an ablative study on the use of $\mathcal{C}_p$ and $\mathcal{C}_d$.
>
> A9: Thanks for your constructive comments. We denote our model without using $\mathcal{C}_p$ and $\mathcal{C}_d$ as Ours-w/oCp and Ours-w/oCd, respectively.
>
> 1. When compared with Ours, the performances of Ours-w/oCp and Ours-w/oCd degrade about 0.9%$\sim$1.5%. It validates the effectiveness of our model to eliminate the noisy pseudo label generation by performing the intersection operation between $\mathcal{C}_p$ and $\mathcal{C}_d$.
>
> -------------------------------------------------------------------------------------------------------------------------------------------------------------
>
> Avg. (AP) $\qquad$ | Office-31 $~$ | Office-Caltech | Office-Home | Digits-Five|
>
> Ours-w/oCp $~~~$ | $\quad$90.5 $\quad~$ | $\qquad$96.9 $\quad~~~$| $\quad$75.1 $\qquad$ | $\quad$93.7 $\quad$ |
>
> Ours-w/oCd $~~~$ | $\quad$90.3 $\quad~$ | $\qquad$97.2 $\quad~~~$| $\quad$75.3 $\qquad$ | $\quad$93.3 $\quad$ |
>
> Ours $\qquad\quad~~$ | $\quad$91.6 $\quad~$ | $\qquad$98.4 $\quad~~~$| $\quad$76.2 $\qquad$ | $\quad$94.6 $\quad$ |
>
> -------------------------------------------------------------------------------------------------------------------------------------------------------------
>
> 2. Please note that our model only using $\mathcal{C}_d$ is still significantly different from the pseudo-labeling in DECISION [1], due to the following reasons.
>
>   The cluster centroid based pseudo-labeling strategy in [1] cannot eliminate the noisy unconfident labels in the generation process, since [1] only depends on nearest distance measure to learn pseudo label. This strategy may result in that the generation process has high probability to obtain noisy labels when the strategy does not match the target data (an unsuitable perspective causes more noisy labels). Different from [1], we generate pseudo labels from two different perspectives, i.e., geometry and probability. Based on these two  perspectives, we develop a confident anchor-induced pseudo label generator to mine confident pseudo labels while addressing noisy unconfident labels. Specifically, we first perform the intersection operation between probability-based confident anchor group $\mathcal{C}_p$ and distance-based confident anchor group $\mathcal{C}_d$ to eliminate the noisy pseudo labels in confident anchor group. Then, we use the adjacent areas of the unconfident sample instead of only the unconfident sample to search a semantic-nearest confident anchor. We believe using the adjacent areas of unconfident samples will be more robust than only using one unconfident sample to search a semantic-nearest confident anchor. Such similarity searching mechanism improves the generalization ability of pseudo label generation while eliminating the noisy similarity matching. We then use the random weight to fuse the unconfident target data and their corresponding confident anchor in the feature space for feature augmentation and pseudo label generation.
>
> Q10: On a side note, it would be a good idea to present standard deviation along with the mean statistics since the average performance of the methods are close.
>
> A10: Thanks for your suggestion. We will present the standard deviation along with the mean statistics in the final revision.

---

> ### Author Response · Authors · 2021-08-26
> **Response to Reviewer R2JJ**
>
> Dear Reviewer R2JJ,
>
> We have tried our best to address all the concerns and provided explanations to all questions. If there are still unclear parts to you, please kindly let us know. We are very glad to further discuss them.
>
> Best,
>
> Authors

---

> > ### Comment · Reviewer_R2JJ · 2021-08-27
> > **Increasing my score**
> >
> > Thanks to the authors for the detailed response. My major concerns are addressed, and I would like to increase the score.
> >
> > A lot of clarifications have been made in the response that I believe can be incorporated into the revision. Particularly, the discussion in A1-3, A8-9 can be added as supplementary material (considering space limitations) with relevant references in the paper to make it more insightful for the reader.

---

### Official Review · Reviewer_4wZc · 2021-07-16

**Rating:** 7
**Confidence:** 4

**Summary:**

This work considers unsupervised domain adaptation issue. A novel framework called CAiDA is proposed which uses the pre-trained source models rather than source data which are commonly used in traditional methods. To demonstrate the effectiveness and superiority of the proposed method, the authors did theoretical analyses as well as numerical experiments. The numerical experimental results show the proposed model can do better than state of art methods.

**Limitations And Societal Impact:**

See Cons above.

**Main Review:**

Pros:
This paper is well organized, the motivation is clear and logical, and it is easy to follow the major idea.

The results are valid. Extensive experimental results are provided to show the effectiveness of the proposed method. Besides, some theoretical analyses are also provided to provide insights behind the proposed model.

Cons:
The writing of this paper can be further polished, especially the theoretical part. The theorems should be stated in a clearer and more formal way. The theorem and assumptions should use the italic form, but the proofs should not. Besides, there are too many steps in the proof of the two theorems, these should be simplified.

There are also some typos in the paper, these should be revised in the final version.

The authors only discussed the limitations about the assumptions in theoretical analysis. The limitations of this model and why it does not work very well on some specific datasets should also be included.


**Time Spent Reviewing:**

2h

---

> ### Author Response · Authors · 2021-08-09
> **Response to Reviewer 4wZc (Q1--Q3)**
>
> Q1: The writing of this paper can be further polished, especially the theoretical part. The theorems should be stated in a clearer and more formal way. The theorem and assumptions should use the italic form, but the proofs should not. Besides, there are too many steps in the proof of the two theorems, these should be simplified.
>
> A1: Thanks for your valuable comments. We will carefully polish this paper, especially the theoretical analysis in the final revision. We will introduce the theorems in a clearer and more formal way, and simplify the proofs of the two theorems as much as possible.
>
> Q2: There are also some typos in the paper, these should be revised in the final version.
>
> A2: Thanks for your suggestion. We will carefully proofread this paper in the final version.
>
> Q3: The authors only discussed the limitations about the assumptions in theoretical analysis. The limitations of this model and why it does not work very well on some specific datasets should also be included.
>
> A3: Thanks for your constructive comments. Our proposed model relies on multiple well pre-trained source models to generate confident pseudo labels for target samples. When multiple source models are not pre-trained well using source data, our model could generate more noisy pseudo labels instead of confident pseudo labels, which heavily degrades the performance of our model on some specific datasets. Moreover, the performance of our model will be limited when the distribution discrepancy between multiple source domains and target domains is intolerably large.

---

> > ### Comment · Reviewer_4wZc · 2021-08-26
> > **question**
> >
> > Thanks for the authors' response. It solved most of my concerns. However, I still wonder something about the theoretical part. You mentioned that your theoretical work can be used to improve other fields (meta-learning, domain generalization). How can this be achieved?

---

> > > ### Author Response · Authors · 2021-08-26
> > > **Response to Reviewer 4wZc**
> > >
> > > Q1: How can our theoretical work be used to improve other fields (meta-learning, domain generalization) ?
> > >
> > > A1: Thanks for your constructive and insightful comments. First, we introduce the difference between our theoretical work and other fields, i.e., meta-learning and domain generalization. Then, we introduce how to improve other fields.
> > >
> > > 1. The theoretical works of meta-learning and domain generalization mainly focus on the generalization of the mean performance of differet target domains. In detail, we denote $\tau$ as the meta distribution and $n$ different domains are $P_1,...,P_n \sim \tau$, i.i.d. The theoretical works of meta-learning and domain generalization, aim to estimate the transfer risk $\mathbf{R}(A,\tau)$ and  $|\mathbb{E}_{P \sim \tau} R(h,P) - \frac{1}{n}  \sum_i \widehat{R}(h,P_i) |$
> > >  Refs[2], respectively, where $\mathbf{R}(A,\tau)$ is the transfer risk defined in Refs[1], $A$ is the meta algorithm and h is some hypothesis functions selected from some hypothesis space.
> > >
> > > 2. According to the introduction mentioned above and previous literature, we discover that almost all theoretical works aim to estimate above mean risk by the number of source domains $P_1,...,P_n \sim \tau$.
> > > However, we note that the mean estimation is not related to a fixed target domain  $P_t$, which is drawn from $\tau$. That is if we give a target domain  $P_t \sim \tau$, previous works cannot estimate
> > > the target risk, i.e., $|R(h,P_t)-\widehat{R}(h, P_t)|$. However, our work can provide the target risk estimation.
> > >
> > > Then, we introduce how to improve.
> > >
> > > 3. According to the idea of Theorem 1 in our paper, we can obtain a result for meta learning and domain generalization:
> > > for any $\epsilon, \delta>0$, we can find a source domain $P_j$ from $P_1,...,P_n$, such that with probability at least $1-\delta-N(\epsilon, P_t)^n>0$, the $L_1$ distance between $P_j$ anf $P_t$ is smaller than $\epsilon$, where $N(\epsilon, P_t)^n$ will be $0$, when $n$ approaches infinity.
> > >
> > > Above results give an estimation about how many source domains can guarantee there exists at least one source domain, which is close to the target domain. This result is first proposed and novel.
> > >
> > > Then, by combining transfer learning theory with above result, we can use the empirical risk minimization strategy to estimate the target risk. The basic idea is similar to Theorem 2 in our paper.
> > >
> > >
> > > Refs[1] Algorithmic Stability and Meta-Learning, JMLR
> > >
> > > Refs[2] Generalizing from Several Related Classification Tasks to a New Unlabeled Sample, NeurIPS.

---

### Official Review · Reviewer_BbBx · 2021-07-16

**Rating:** 5
**Confidence:** 3

**Summary:**

This paper addresses the multi-source free domain adaptation problem where multiple pretrained source models and unlabeled target data are given. Theoretical analysis aims at proving that multiple source domains improve the probability to ensure a generalization bound. The method is developed based on mutual information maximization, pseudo labeling and consistency losses. Experiments show that the proposed method achieves slightly better results.

**Limitations And Societal Impact:**

see main review

**Main Review:**

This paper addresses a new problem but with limited technical novelty. The proposed method combines a number of well-known techniques. The two loss terms in Eq.(2) and (3) in Section 5.1 form components of the standard mutual information criterion used for semi-supervised learning but without providing the similar insights.  The linear combination of multiple source models is also a very common integration strategy.  The pseudo-label generator in Section 5.2 follows the most common confident-based pseudo label determination strategy but with a bit tedious process. It lacks any more insightful principle.

Moreover, the techniques adopted are mainly semi-supervised learning techniques and there is not really any domain adaptation factor. Does this mean source free domain adaptation can only be solved from the semi-supervised learning perspective?


The related work section is insufficient. It only very briefly mentions the source-free domain adaptation methods. Are there any major technical differences between this proposed method and the methods in the literature? It fails to mention and compare their method with the existing work [1] for the same multi-source free domain adaptation problem in the related work section even though this work [1] also uses information maximization, prototype-base pseudo labeling and meta-learning to learn the weighting coefficients for multiple source domains.

The experiments are not very convincing with slight performance gains. Are all the comparison methods used the same feature extraction networks?  The feature extraction network can lead to performance differences.

Other questions:
1. For the Source-Specific Transferability Perception, in line 180, why are the weight vectors taken as prototypes if the classifiers compute inner-product instead of Euclidean distance?


[1] Sk Miraj Ahmed, et al. Unsupervised multi-source domain adaptation without access to source data. CVPR 2021.


**Time Spent Reviewing:**

x

---

> ### Author Response · Authors · 2021-08-09
> **Response to Reviewer BbBx (Q1--Q4)**
>
> Q1: Novelty. The two loss terms in Eq.(2) and (3) in Section 5.1 form components of the standard mutual information criterion used for semi-supervised learning but without providing the similar insights.
>
> A1: Many thanks for analyzing our model. The main contribution in Section 5.1 of our work is the proposed source-specific transferability perception module instead of Eqs. (2) and (3),  although Eqs. (2) and (3) are two commonly-used loss terms to tackle non-adversarial learning based source free domain adaptation, as introduced in [1][26]. This source-specific transferability perception module is developed to automatically quantify the transferability contributions of complementary knowledge from source domains without access to the source data. With the quantified transferability contribution for each source domain, we extend the standard mutual information criterion used in single source free domain adaptation [26] to multi source free domain adaptation.
>
> Q2: Novelty. The linear combination of multiple source models is also a very common integration strategy.
>
> A2: Thanks for this comment. When using source data to quantify the contributions of source domains, the linear combination of multiple predictions from source domains is a common integration strategy indeed in multi-source domain adaptation [23][30][49][51][55]. However, in the multi-source free domain adaptation field, it is still a challenging task to obtain accurate contribution weight for each source domain without access to the source data. Only one paper DECESION [1] is proposed to tackle this challenge, but [1] heavily depends on the parameter initialization and human prior in the source-free domain adaptation task. To this end, we develop a source-specific transferability perception module in this paper to automatically quantify the transferability contributions of complementary knowledge from source domains, which is also one of our main contributions. Different from existing methods, it can be initialized with random parameters, and learns the transferability contribution weight of each source domain by the network itself and without manual interference (few human prior).
>
> Q3: Novelty. The pseudo-label generator in Section 5.2 follows the most common confident-based pseudo label determination strategy but with a bit tedious process. It lacks any more insightful principle.
>
> A3: Many thanks for analyzing the pseudo label generator. The proposed confident anchor-induced pseudo label generator is one of the core contributions in this paper, which is significantly different with existing methods in both theoretical analysis and methodology aspects:
>
> 1. Theoretical Analysis: To the best of our knowledge, this paper is the first exploration to propose the theoretical analysis about confident pseudo label generation in transfer learning field. Specifically, theorem 1 has shown that more source domains imply more confident pseudo labels under the confident-based pseudo label strategy (anchor point assumption). Our theorem 1 is the first theorem to explore the relationship between multiple source domains and pseudo labels by providing a lower bound for the number of confident pseudo labels. As we known, though many works have studied the confident-based pseudo label strategy, no work has provided the lower bound for the number of the confident pseudo labels. This is our core theoretical contribution. In addition, the math skills developed in theorem 1 can also be extended into other fields, such as meta learning theory, domain generalization theory and multiple source domain adaptation theory.
>
> 2. Methodology: The noisy unconfident label is the key challenge in pseudo label generation process, and our proposed confident anchor-induced pseudo label generator could effectively address it from the following aspects:
>
>  (1) Different from the existing confidence-based or probability-based pseudo label strategies, we develop a distance-based pseudo label generation strategy to construct distance-based confident anchor group $\mathcal{C}_d$. The intersection operation between probability-based confident anchor group $\mathcal{C}_p$ and distance-based confident anchor group $\mathcal{C}_d$ can effectively eliminate the noisy pseudo labels in the confident anchor group.
>
>  (2) To generate confident pseudo labels for unconfident target data, we select a semantic-nearest confident anchor from $\mathcal{C}$ for each target data via continual similarity searching. Specifically, we search a serial of unconfident guiding data consecutively using the distance measure function in the feature space, until a confident anchor from $\mathcal{C}$ is detected. Namely, we consider the similarity relationship between a confident anchor and a local unconfident group including multiple semantically-adjacent unconfident samples. It can also be seen that we use the adjacent areas of this unconfident sample instead of only this unconfident sample to search a semantic-nearest confident anchor. We believe using the adjacent areas of unconfident samples will be more robust than only using one unconfident sample to search a semantic-nearest confident anchor. Such similarity searching mechanism improves the robustness ability to search a semantic-nearest confident anchor while eliminating the noisy similarity matching.
>
>  (3) Different from using the nearest distance alignment to generate pseudo labels for target data, we use the random weight to fuse the unconfident target data and their corresponding confident anchor in the feature space for feature augmentation. The synthetic feature is utilized to perform pseudo label generation for target data. Such feature augmentation strategy compensates the negative influence of noise label in feature space.
>
> Q4: The techniques adopted are mainly semi-supervised learning techniques and there is not really any domain adaptation factor. Does this mean source free domain adaptation can only be solved from the semi-supervised learning perspective?
>
> A4: Thanks for your constructive comments. Although some strategies in this paper are related to semi-supervised learning, the proposed model is highly-related to domain adaptation works. Moreover, the source free domain adaptation cannot only be solved by the semi-supervised learning due to the following reasons.
>
> 1. In semi-supervised learning, little labeled data is available, and there is no large distribution discrepancy between training and test datasets. However, in the setting of multiple source free domain adaptation, we have no any labeled target data, and there is large distribution discrepancy between multiple source domains and target domain. Hence, in some cases, our setting is more challenging than semi-supervised learning, which implies that the pseudo-label strategies used in semi-supervised learning may not work very well in multiple source free domain adaptation. To this end, we develop a confident anchor-induced pseudo label generator to mine confident pseudo labels by considering the distribution discrepancy from multiple source domains.
>
> 2. The proposed source-specific transferability perception relies on the source class-wise prototypes to automatically quantify the transferability contributions of complementary knowledge from source domains. Such weighting strategy has generalization performance for random initialization of network parameters. It is one of main contributions for multi-source free domain adaptation, while having little relation with semi-supervised learning. With the weighting strategy, we aim to drive source models and the target data to match better. The idea to match source and target domains is from domain adaptation.
>
> 3. As we known, there is a method (i.e., Refs[1]) that solves source free problem by combining with existing unsupervised domain adaptation methods. This method aims to replay source data using source model, then uses the generated source data to align distribution gap across domains via existing unsupervised domain adaptation methods. Hence, only semi-supervised learning cannot solve source free domain adaptation well.
>
> Refs[1]: Source-Free Domain Adaptation for Semantic Segmentation, CVPR 2021

---

> ### Author Response · Authors · 2021-08-09
> **Response to Reviewer BbBx (Q5--Q7)**
>
> Q5: The related work section is insufficient. It only very briefly mentions the source-free domain adaptation methods. Are there any major technical differences between this proposed method and the methods in the literature? It fails to mention and compare their method with the existing work [1] for the same multi-source free domain adaptation problem in the related work section though this work [1] also uses information maximization, prototype-base pseudo labeling and meta-learning to learn the weighting coefficients for multiple source domains.
>
> A5: Thanks for your insightful comments. We will carefully polish the related work in the finial revision, and there are several significant differences between our model and [1]:
>
> 1. The pseudo labeling strategy in [1] utilizes the weighted k-means clustering to compute class-wise prototypes, and measures nearest distance between the given sample and class-wise prototypes to generate pseudo label for target sample. However, [1] generates pseudo labels only depending on nearest distance measure. This strategy may result in that the generation process has high probability to generate noisy labels when this strategy does not match the target data (an unsuitable perspective causes more noisy labels). Different from [1], we generate pseudo labels from two different perspectives, i.e., geometry and probability. Based on these two perspectives, we develop the confident anchor-induced pseudo label generator to mine confident pseudo labels while addressing noisy unconfident labels. Specifically, we first perform the intersection operation between probability-based confident anchor group $\mathcal{C}_p$ and distance-based confident anchor group $\mathcal{C}_d$ to eliminate the noisy pseudo labels in confident anchor group. Then, we use the adjacent areas of the unconfident sample instead of only the unconfident sample to search a semantic-nearest confident anchor. Such similarity searching mechanism improves the generalization ability of pseudo label generation while eliminating the noisy similarity matching. Meanwhile, we use the random weight to fuse the unconfident target data and their corresponding confident anchor in the feature space for feature augmentation and pseudo label generation.
>
> 2. The source-weighting strategy in [1] heavily relies on the parameter initialization and human prior. Different from [1], the source-specific transferability perception module developed in our work can automatically quantify the transferability contributions of complementary knowledge from source domains. This strategy can be initialized with random parameters, and learns the transferability contribution weight of each source domain by the network itself and without manual interference (few human prior).
>
> 3. Different from [1], we also design a class relationship-aware consistency loss to encourage target data from the same class to be compactly clustered together, while preserving the intrinsic inter-class relationships via soft confusion matrix alignment.
>
> 4. Compared with [1], our theoretical analysis is the first exploration to illustrate the relationship between multiple sources and pseudo labels by providing a lower bound for the number of confident pseudo labels. As we known, although many works have studied the confident-based pseudo label strategy, no other work has provided the lower bound for the number of the confident pseudo labels.
>
> Q6. Are all the comparison methods used the same feature extraction networks? The feature extraction network can lead to performance differences. The experiments are not very convincing with slight performance gains.
>
>  A6. Thanks for your constructive comments.
>
>  1. For a fair comparison, all the competing methods use the same feature extraction network.
>
>  2. Our proposed model achieves slight performance improvement on Office datasets, since these Office datasets are not challenging enough for recent domain adaptation works and the competing methods are easy to overfit. To further validate the effectiveness of our model, we conduct comparison experiments on the challenging person re-identification tasks by using Market1501 (Ma), DukeMTMCreID (Du), CUHK03 (Cu) and MSMT (Ms) datasets, and set same experimental configurations with Refs[2]. From the following results, we can observe that our model achieves significant performance improvement than multi-source domain adaptation methods (Refs[1] and Refs [2]) with access to source data. Moreover, when compared with source free domain adaptation methods (DECISION [1], SHOT [26] and USFDA Refs[3]), our model also significantly outperforms them by a large margin. It validates the effectiveness of our model to perform the challenging multi-source free domain adaptation tasks.
>
> --------------------------------------------------------------------------------------------------------------------------------------------
>
> Metric (mAP // R-1 // R-5) | $~~$ Du+Cu => Ma $\quad$ |  Du+Cu+Ms => Ma $~$| $~~~$ Ma+Cu => Du $~~~$ |
>
> MMT(DBSCAN)   Refs[1]    $~~~$   | 75.3 // 89.5 // 96.6  | 74.8  // 89.3  //  96.2  |  65.7 // 79.0 // 89.0 |
>
> MDIF+RDSBN Refs[2] $\quad~$ | 85.2 // 94.2 // 98.0 | 86.0 // 94.8 // 97.9 | 69.0 // 81.2 // 90.3 |
>
> SHOT [26] $\qquad\qquad\quad~~$ | 81.6 // 92.1 // 94.3 | 81.9 // 91.7 // 92.8 | 64.4 // 76.6 // 84.1 |
>
> USFDA  Refs[3] $\qquad\quad~~~$ | 79.3 // 91.3 // 93.2 | 80.1 // 90.5 // 90.8 | 62.8 // 73.4 // 82.5 |
>
> DECISION [1] $\qquad\qquad~$ | 83.8 // 92.6 // 96.5 | 83.6 // 92.7 // 95.4 | 67.3 // 78.9 // 88.6 |
>
> Ours $\qquad\qquad\qquad\quad~~$ | 86.4 // 95.0 // 98.7 | 87.3 // 96.1 // 98.5 | 69.8 // 82.1 // 91.3 |
>
> ---------------------------------------------------------------------------------------------------------------------------------------
>
> Refs[1] Mutual meanteaching: Pseudo label refinery for unsupervised domain adaptation on person re-identification, ICML 2020
>
>  Refs[2] Unsupervised Multi-Source Domain Adaptation for Person Re-Identification, CVPR 2021
>
>  Refs[3] Universal Source-Free Domain Adaptation, CVPR 2020
>
> Q7: For the Source-Specific Transferability Perception, in line 180, why are the weight vectors taken as prototypes if the classifiers compute inner-product instead of Euclidean distance?
>
> A7: Thanks for your insightful comments. We experimentally find that the weight matrix of the last fully connected layer in the classifier of source model is more discriminative to characterize the prototypes, when having no acces to the source data. It can be seen that the prototypes can be better distinguished in this more discriminative space.

---

> ### Author Response · Authors · 2021-08-26
> **Response to Reviewer BbBx**
>
> Dear Reviewer BbBx,
>
> We have tried our best to address all the concerns and provided explanations to all questions. If there are still unclear parts to you, please kindly let us know. We are very glad to further discuss them.
>
> Best,
>
> Authors

---

### Decision · Program_Chairs · 2021-09-27

**Decision:**

Accept (Poster)

**Comment:**

The theoretical contribution for multi-source domain adaptation without source data is nice. Though some components in the proposed method are borrowed from existing techniques, the overall design of the proposed method is based on the insights of the theoretical analysis. As [1] is the first work (if I am not wrong), which only provides theoretical analysis, the authors should add a section to discuss the difference in terms of the theoretical findings between the proposed method and [1] in a revision.

Overall, I recommend acceptance for this work.